

# Observations of particle number size distributions and new particle formation in six Indian locations

Mathew Sebastian[1], Sobhan Kumar Kompalli[2], Anil Kumar[3], Sandhya Jose[4,5], S. Suresh Babu[2], Govindan Pandithurai[3], Sachidanand Singh[4,5], Rakesh K. Hooda[6], Vijay K. Soni[7], Jeffrey R. Pierce[8], Ville Vakkari[6,9], Eija Asmi[6], Daniel M. Westervelt[10,11], Antti-P. Hyvärinen[6], and Vijay P. Kanawade[1,*]

[1]Centre for Earth, Ocean and Atmospheric Sciences, University of Hyderabad, Hyderabad, India
[2]Space Physics Laboratory, Vikram Sarabhai Space Centre, Thiruvananthapuram, India
[3]Indian Institute of Tropical Meteorology, Ministry of Earth Sciences, Pune, India
[4]CSIR-National Physical Laboratory, Dr. K.S. Krishnan Road, New Delhi, India
[5]Academy of Scientific and Innovative Research (AcSIR), Ghaziabad-201002, India
[6]Finnish Meteorological Institute, Erik Palmenin Aukio 1, Helsinki, Finland
[7]India Meteorological Department, Ministry of Earth Sciences, New Delhi, India
[8]Department of Atmospheric Science, Colorado State University, Fort Collins, CO, USA
[9]Atmospheric Chemistry Research Group, Chemical Resource Beneficiation, North-West University, Potchefstroom, South Africa
[10]Lamont-Doherty Earth Observatory of Columbia University, New York, USA
[11]NASA Goddard Institute for Space Studies, New York, NY, USA

Correspondence to: Vijay P. Kanawade (vijaykanawade03@yahoo.co.in)

**Abstract.** Atmospheric new particle formation (NPF) is a crucial process driving aerosol number concentrations in the atmosphere; it can significantly impact the evolution of atmospheric aerosol and cloud processes. This study analyses at least one year of asynchronous particle number size distributions at six different locations in India. We also analyze the frequency of NPF and its contribution to cloud condensation nuclei (CCN) concentrations. We found that the NPF frequency has a considerable seasonal variability. At the measurement sites analyzed in this study, NPF frequently occurs in March-May (pre-monsoon, about 21% of the days) and is the least common in October-November (post-monsoon, about 7% of the days). Considering the NPF events in all locations, the particle formation rate ($J_{NUC}$) varied by more than an order of magnitude (0.01 - 0.6





cm$^{-3}$ s$^{-1}$) and the growth rate (GR$_{NUC}$) by about three orders of magnitude (0.2 - 17.2 nm h$^{-1}$). We found that J$_{NUC}$ was higher by nearly an order of magnitude during NPF events in urban areas than mountain sites. GR$_{NUC}$ did not show a systematic difference. Our results showed that NPF events could significantly modulate the shape of particle number size distributions and CCN concentrations in India. The contribution of a given NPF event to CCN concentrations was the highest in urban locations (4.3×10$^3$ cm$^{-3}$ per event and 1.2×10$^3$ cm$^{-3}$ per event for 50 nm and 100 nm, respectively) as compared to mountain-background sites (2.7×10$^3$ cm$^{-3}$ per event and 1.0×10$^3$ cm$^{-3}$ per event). To better understand atmospheric NPF and its contribution to CCN concentrations, we would need long-term observational data from various diverse environments in India, aided with regional model simulations to help interpret field observations.

**Keywords:** new particle formation, particle number size distribution, Aitken mode, accumulation mode, cloud condensation nuclei

## 1 Introduction

Cooling by atmospheric aerosols offset a significant fraction of the radiative forcing of the greenhouse gases (Paasonen et al., 2013) directly by scattering and absorbing solar radiation and indirectly by altering cloud microphysical properties via activation of cloud condensation nuclei (CCN) (Rosenfeld et al., 2014; Sarangi et al., 2018). New particle formation (NPF), as a result of the gas-to-particle conversion, is the largest source of the aerosol number to the terrestrial atmosphere (Kulmala et al., 2007; Zhang et al., 2012). While nucleated particles from NPF are initially very small molecular clusters (1-2 nm; Kerminen et al., 2012), these molecular clusters can grow to large sizes within a few hours to a few days and ultimately reach CCN-active sizes (>50-100 nm) (Pierce and Adams, 2007; Westervelt et al., 2013). Thus, CCN forms the direct microphysical link between aerosols and clouds and plays a vital role in the hydrological cycle and climate.

In India, several intensive field campaigns such as the Indian Ocean Experiment (INDOEX) (Ramanathan et al., 2001), Indian Space Research Organization (ISRO)-Geosphere-Biosphere Programme (GBP)- Land campaign II (Tripathi et al., 2006; Tare et al., 2006), and Integrated Campaign for Aerosols, gases, and Radiation Budget (ICARB) (Moorthy et al., 2008; Nair et al., 2020; Kompalli et al., 2020) measured sub-micron particle number size distributions



(PNSDs). There are also short- and long-term field observations of sub-micron PNSDs in a variety
of diverse locations in India (Hyvärinen et al., 2010; Kanawade et al., 2014a; Shika et al., 2020;
Tripathi et al., 1988; Komppula et al., 2009; Singh et al., 2004; Moorthy et al., 2011; Babu et al.,
2016; Kompalli et al., 2018). But there are sparse studies in India characterizing seasonal variation
in PNSDs and number concentrations (Kanawade et al., 2014a; Hyvärinen et al., 2010; Komppula
et al., 2009; Hooda et al., 2018; Laj et al., 2020) and atmospheric NPF (Sebastian et al., 2021b;
Siingh et al., 2018; Neitola et al., 2011; Moorthy et al., 2011; Kanawade et al., 2014b; Kanawade
et al., 2014c; Kanawade et al., 2020a). The characterization of PNSDs is critical because the PNSD
is controlled by an evolving balance between NPF, condensation of vapor on pre-existing particles,
evaporation of particles, coagulation and sedimentation (Ipcc, 2013). Previous field measurements
and modeling studies globally demonstrated a substantial enhancement in CCN number
concentrations from nucleation (Yu et al., 2020; Wiedensohler et al., 2009; Sihto et al., 2011; Rose
et al., 2017; Tröstl et al., 2016; Kalivitis et al., 2015; Westervelt et al., 2013; Pierce et al., 2012;
Pierce et al., 2014; Westervelt et al., 2014; Kerminen et al., 2012; Kerminen et al., 2018; Merikanto
et al., 2009; Gordon et al., 2017). For instance, Merikanto et al. (2009) revealed that 45% of the
global low-level CCN at 0.2% supersaturation originates from nucleation. Westervelt et al. (2014)
also found that nucleation contributes to about half of the boundary layer CCN (at supersaturation
of 0.2%), with an estimated uncertainty range of 49 to 78%, which is sensitive to the choice of
nucleation scheme. In contrast, Reddington et al. (2011), using the global model GLOMAP against
ground-based measurements at 15 European sites, found that CCN-sized particle number
concentrations were driven by processes other than nucleation at more than ten sites. They
explained that the weakened response of CCN-sized particles to boundary layer nucleation arises
from an increase in coagulation and condensation sinks for ultrafine particles, thereby reducing
the condensational growth of ultrafine particles to CCN-active sizes (Kuang et al., 2009; Pierce
and Adams, 2007). Tröstl et al. (2016) also revealed that only a small fraction of total particles less
than 50 nm grew beyond 90 nm (50-100 particles cm$^{-3}$), even on a timescale of several days.
Therefore, to better understand atmospheric NPF and its contribution to the boundary layer CCN
budget, we need highly-resolved spatiotemporal observational data in diverse environments
globally, aided with aerosol model simulations, to help to interpret field observations.

Overall, studies pertinent to the impact of NPF on aerosol-cloud interactions are highly

sparse in India. The sources of aerosols, and gaseous precursors required for secondary aerosol





formation, depict a considerable spatiotemporal heterogeneity over India. Therefore, observational
aerosols and precursors data must be synthesized to understand the processes that govern NPF and
its contribution to CCN concentrations in different settings of India. The primary objective of this
study is to harmonize observational PNSDs data from six diverse locations in India to present
analyses of PNSDs, atmospheric NPF, and the contribution of NPF to CCN concentrations.

**2 Methods**
**2.1 Observation sites and aerosol sampling instrumentation**
Figure 1 shows the geographical location of measurement sites on the surface elevation
map. Table 1 provides details of measurement sites and particle data analyzed in this study.

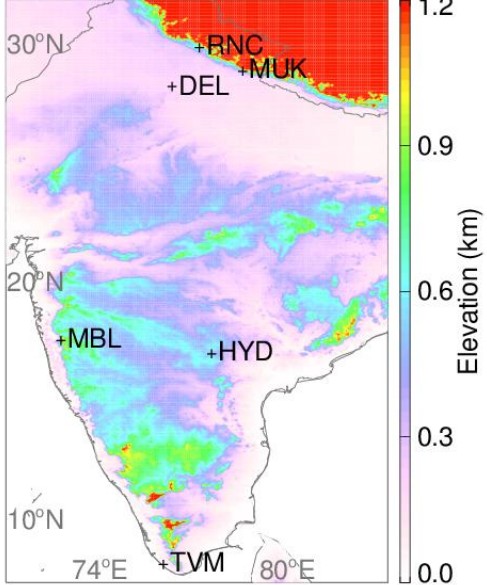


**Figure 1.** The geographical location of measurement sites on the surface elevation map.
Measurement sites such as Ranichauri (RNC), Mukteshwar (MUK), Mahabaleshwar (MBL),
Hyderabad (HYD), Thiruvananthapuram (TVM), and Delhi (DEL) are shown by the plus sign.
The global 1-arcsecond (30-m) SRTM digital surface elevation data is obtained from the United
States Geological Survey (https://dds.cr.usgs.gov/srtm/version2_1/SRTM30/).
**Table 1.** Details of the measurement sites and particle number size distribution measurements
analyzed in this study.

| Site Name | Site code | Site type | Instrument | Size range (nm) | Time resolution (minutes) | Time Period |
|---|---|---|---|---|---|---|
| Ranichauri | RNC | Mountain background | DMPS | 10.1–757 | 10 | 12/2016 – 09/2018 |
| Mukteshwar | MUK | Mountain background | DMPS | 10.1–757 | 5 | 01/2012 – 12/2013 |
| Mahabaleshwar | MBL | Mountain semi-rural | WRAS | 5.14–1000 | 4 | 03/2015 – 03/2016 |
| Hyderabad | HYD | Urban | SMPS | 10.9–514 | 5 | 04/2019 – 03/2020 |
| Thiruvananthapuram | TVM | Semi-urban coastal | SMPS | 14.6–661.2 | 5 | 01/2013 – 01/2014 |
| Delhi | DEL | Urban | WRAS | 5.14–1000 | 5 | 11/2011 – 01/2013 |

DMPS: Differential Mobility Particle Sizer, WRAS: Wide-Range Aerosol Spectrometer, SMPS:
Scanning Mobility Particle Sizer

Ranichauri observation site (RNC, 30.2° N, 78.25° E; ~1930 m above mean sea level, amsl)
is located in Tehri–Garhwal district of Uttarakhand state in the southern slope of the Western
Himalaya. The RNC site is situated on an isolated hilltop within the campus of the College of
Forestry in the Ranichauri village. The RNC site is a Climate Monitoring station managed by the
India Meteorological Department (IMD). It is a mountain background remote observatory
(Sebastian et al., 2021b) and located about 70 km to the northeast of Rishikesh city, about 100 km
to the northwest of the Srinagar city, and about 100 km to the east of Dehradun. Here, particle
number size distributions in the size range from 10 nm to 757 nm (30 size bins) is measured using
a differential mobility particle sizer (DMPS, Finnish Meteorological Institute assembled) from
December 2016 – September 2018 are used (Sebastian et al., 2021b). The DMPS consisted of a
Vienna-type differential mobility analyzer (DMA) that classifies the charged particles according
to their electrical mobility and a TSI 3772 condensation particle counter (CPC) that counts
particles of the selected mobility. The sample air was drawn inside through a stainless-steel inlet
tube of about 2 meters in length and dried to less than 40% relative humidity with a Nafion dryer
(Perma Pure model MD-700-48). Diffusion losses in the inlet and inside the DMPS instrument
were considered in the data inversion. The inversion method was identical to that presented by
Wiedensohler et al. (2012) for the Finnish Meteorological Institute (FMI) DMPS.





Mukteshwar observation site (MUK, 29.43° N, 79.62° E, 2180 m amsl) is located in the
Nainital district of Uttarakhand state in the southern slope of the Central Himalaya. The
Mukteshwar village is situated 3 km to the northeast of the measurement site at a similar altitude
with ~800 inhabitants (Census of India, 2011). MUK can be considered a mountain background
site, with the annual mean black carbon (BC) concentration of 0.9 µg m$^{-3}$. The town of Almora
(1650 m amsl, 34,000 inhabitants) is located at about 16 km to the north, Nainital (1960 m amsl,
41000 inhabitants) is located at about 25 km to the southwest, and the city of Haldwani (424 m
amsl, 150,000 inhabitants) is located at about 32 km to the southwest to MUK. Delhi, the major
metropolitan city (215 m amsl, 16.8 million inhabitants), is located approximately 250 km to the
southwest. Systematic measurements of aerosol properties have been conducted at MUK since
2005 in Indo-Finnish cooperation with the Finnish Meteorological Institute (Hooda et al., 2018
and references therein). Here, we used only two years (January 2012 to December 2013) of
measurements of particle number size distributions in the size range of 10 nm to 757 nm (30 size
bins). The air sampling procedure was similar to that of the RNC observation site.
Delhi observation site (DEL, 28.64° N, 77.17° E, 215 m amsl) is located at CSIR-National
Physical Laboratory (NPL). Delhi, India's national capital and largest metropolitan city in South
Asia, is located in the northwestern Indo Gangetic Plain (IGP) in northern India. Delhi city has a
population of 16.8 million, with a population density of 11,320 km$^{-2}$ (Census of India, 2011). The
Great Indian Desert (Thar Desert) of Rajasthan state is located to the southwest, hot central plains
to the south, and hilly regions to the north and east of Delhi. Long-range transported air masses
often influence Delhi's air quality from the northwest (agricultural residue burning from Punjab
and Haryana in October-November) and southwest (dust storms from Thar and Arabian Peninsula
in April-June) (Kanawade et al., 2020b; Srivastava et al., 2014). Wide Range Aerosol
Spectrometer (WRAS, manufactured by GRIMM, Germany), installed on the second floor of the
NPL main building, was used to measure particle number size distributions. WRAS consists of a
Scanning Mobility Particle Sizer (SMPS) and an Environmental Dust Monitor (EDM). GRIMM-
SMPS system consists of a Vienna-type monodisperse differential mobility analyzer (M-DMA).
DMA classifies the particle according to their electrical mobility, which is then counted using a
CPC. EDM uses an Optical Particle Counter (OPC), which works on the light scattering
technology for particle counting gives the particle number size distribution in the size range from
250 nm to 32 µm (Grimm and Eatough, 2009). Thus, the WRAS system gives the particle number





size distribution in the size range from 5.5 nm to 32 μm (72 size bins). The detailed description
and principle of the instrument are discussed elsewhere (Grimm and Eatough, 2009). In this study,
we used particle number size distributions in the size range of 5.14 nm to 1000 nm from November
2011 to January 2013.

Mahabaleshwar observation site (MBL, 17.92° N, 73.65° E; 1378 m amsl) is located in the

small town named Mahabaleshwar in the forested Western Ghats range in the Satara district of
Maharashtra State. In MBL, a High-Altitude Cloud Physics Laboratory (HACPL) was established
by the Indian Institute of Tropical Meteorology (IITM), Pune, in 2012, to study monsoon clouds
in this region. HACPL site details are found elsewhere (Anil Kumar et al., 2021). Mahabaleshwar
town is a tourist attraction consisting of dense vegetation, residential houses, hotels, and a rural
market. Pune city is located on the leeward side of the Western Ghats about 100 km to the north,
Mumbai city is located approximately 250 km on the northwest, and Satara city is located
approximately 50 km to the southeast of Mahabaleshwar. Measurements of particle number size
distributions were carried out using the GRIMM-WRAS system. The detailed description and
principle of the instrument are discussed elsewhere (Grimm and Eatough, 2009). The sampling
probe uses a Nafion dryer to reduce the relative humidity to ~40%. In this study, we used particle
number size distributions in the size range of 5.14 nm to 1000 nm from March 2015 to March

2016.

Hyderabad observation site (HYD, 17.46° N, 78.32° E; 542 m amsl), University of

Hyderabad, is located in the outskirts of Hyderabad urban city. HYD observation site details can
be found in Sebastian et al. (2021a). Briefly, particle number size distributions in size range from
10.9 to 514 nm (108 size bins) were measured using TSI SMPS, which consists of an electrostatic
classifier with a long differential mobility analyzer (TSI LDMA, model 3082) and a butanol CPC
(TSI, model 3772), on the second floor of the Earth Sciences building located in the University of
Hyderabad campus from April 2019 to March 2020. The scanning cycle of SMPS was 300
seconds, yielding a particle number size distribution every 5 minutes.

Thiruvananthapuram (Trivandrum) observation site (TVM, 8.55° N, 76.97°E, 3 m amsl) is

a tropical semi-urban coastal city with a population of ~1 million (Census of India, 2011), located
on the southwestern coast of the Indian peninsular. The observations were carried at the Space
Physics Laboratory (SPL) within the Thumba Equatorial Rocket Launching Station, which is about
500 m due east of the Arabian Sea coast and 10 km northwest of the urban area of



Thiruvananthapuram.  The experimental site is free from major industrial or urban activities (Babu
et al., 2016). TVM station is a part of the Aerosol Radiative Forcing over India (ARFI) project
network of the Indian Space Research Organisation - Geosphere-Biosphere Program (ISRO-GBP).
Measurements of particle number size distributions in size range from 14.6 nm to 661.2 nm (108
size bins) were made using TSI SMPS, which consists of an electrostatic classifier with an LDMA
(3081) and a water-based CPC (3786) from January 2013 to January 2014. More details about the
site and prevailing meteorology are described in Babu et al. (2016).

Particle number size distributions are categorized by season. We have defined four seasons

as indicated in Table 2. The overall particle number size distribution data coverage was adequate
(>60 %) at the RNC, MUK, MBL, and HYD sites (Fig. 2) for determining the main seasonal and
annual features of particle number size distributions and NPF characteristics. The data coverage at
TVM (34%) and DEL (47%) was lower. We also analyzed the number concentration of three sub-
micron aerosol modes: Aitken mode (25-100 nm), accumulation mode (100-514 nm), and total
particles (<514 nm).

**Table 2**. Seasons are defined in the analysis and average weather conditions.

| Season | Months | Comments |
|---|---|---|
| Winter | December, January, February | Cold and dry |
| Pre-monsoon | March, April, May | Hot and dry |
| Monsoon | June, July, August, September | Warm, humid, and wet |
| Post-monsoon | October, November | Cool and humid |




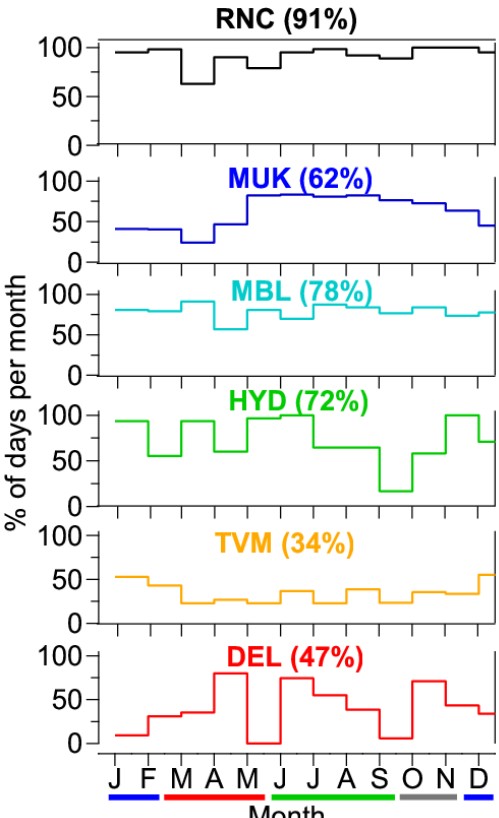


**Figure 2**. Particle number size distributions data coverage (% of days/month) at the sites. The values in the bracket indicate total data coverage. The blue, red, green, and grey colored thick lines indicate winter, pre-monsoon, monsoon, and post-monsoon months.


**2.2 New particle formation event classification and features**

We classified observation days into three types of events: NPF event day, non-event day, and undefined event day using visual inspection of the particle number size distributions following the methodology given by Dal Maso et al. (2005). A day was classified as an NPF event day by the presence of a distinctly new mode of particles with a diameter smaller than 25 nm and steady growth in diameter of this new mode such that the particle number size distributions display a noontime "banana" shaped aerosol growth. The particle mode diameter (i.e., the local maximum of the particle number size distribution) was obtained by fitting a log-normal distribution to the measured particle number size distribution. A day without any evidence of a distinctly new mode





of particles diameter smaller than 25 nm was identified as a non-event day. Those days, which
were difficult to be classified as any one of the above two event types, were identified as undefined
event days. For NPF events, the particle growth rate (GR) was calculated by fitting a first-order
polynomial line through growing particle mode diameter between the lowest detectable size (LDS)
of the instrument (e.g., 10 nm for RNC) and 25 nm as a function of time and calculating its slope.
The formation rate of a particle at the LDS ($J_{LDS}$) was also found using the simplified
approximation of the General Dynamic Equation (GDE), describing the evolution of the particle
number size distribution as given below;

$$J_{LDS} = \frac{dN_{LDS-25}}{dt} + F_{CoagS} + F_{growth} \tag{1}$$

where the first term in Eq. (1) is the rate of the change of nucleation mode particle number
concentrations, the second term is the coagulation loss of nucleation mode particles, and the third
term is the flux out of the size range of LDS-25 nm, i.e., condensational growth (Dal Maso et al.,
2005). A direct comparison of GR and J between all of the sites is not possible because of the
different size ranges covered by the instruments.

**2.3 Increase in CCN concentrations from NPF**
The increase in CCN concentrations from any given NPF event can be estimated by
comparing the CCN concentration before the event ($N_{CCNprior}$) and the maximum CCN
concentration during the event ($N_{CCNmax}$) following the methodology developed by Kerminen et
al. (2012), which we modified further. In typical ambient in-cloud supersaturations, the total
number of particles from 50 nm to >100 nm can be considered as a proxy for CCN concentrations
(Westervelt et al., 2013; Kerminen et al., 2012). $N_{CCNprior}$ was chosen to be a one-hour average
concentration of particles larger than 50 nm (and 100 nm) just before the start of the NPF event.
$N_{CCNmax}$ was taken as a maximum one-hour average concentration of particles larger than 50 nm
(and 100 nm) during the event. The $N_{CCNmax}$ is not the best representation of CCN concentration
after the NPF event because it is not possible to estimate the end of an NPF event. But it gives a
rough estimate of the observed maximum number of primary and secondary particles present in
the atmosphere during an event (Kerminen et al., 2012). We calculated the seasonally averaged
change in CCN-active particles on non-event days over the same time of day as the NPF events,





which would account for the CCN concentrations from processes other than NPF. Then, the
absolute increase in CCN concentration from NPF is calculated as given below,

CCN increase = $(N_{CCNmax} - N_{CCNprior})_{NPFevent} - (N_{CCNmax} - N_{CCNprior})_{non-events}$        (2)

The first term on the right-hand side in Eq. (2) indicates the CCN increase during an NPF event,
while the second term indicates the CCN increase during a non-event. This difference between
them allows us distinguishing primary particles and particles formed originally from atmospheric
nucleation and yields the best representation of CCN concentrations after the NPF event. But the
atmospheric condition on non-event days is generally different from NPF event days; therefore,
the calculated increase in CCN concentrations from NPF may be imprecise.

**3. Results and discussion**
**3.1 Variability in particle number size distributions and number concentrations**

Figure 3 shows the annual and seasonal median and 25th and 75th percentile values of

particle number size distributions at all the sites. The thick line represents the median value,
whereas the shaded area indicates particle number size distribution between 25th and 75th
percentiles. The annual median particle number size distribution has the smallest mode diameter
at DEL compared to the other sites. The smallest mode diameter necessarily indicates the
significant near-surface anthropogenic sources at DEL as compared to other sites. The mountain
sites (RNC, MUK, and MBL) all show similar mode diameters, with the lowest concentrations at
RNC. Amongst urban areas (HYD, TVM, and DEL), TVM has the largest mode diameter, which
is frequently influenced by the influx of marine air masses containing high moisture and coarser
sea salt aerosols (Babu et al., 2016) (Fig. 3a). The peak number concentration of PNSDs is the
highest in pre-monsoon (MAM) than in other seasons at RNC and MUK (Fig. 3b-c), while it was
similar in winter and pre-monsoon at MBL (Fig. 3d). These elevated concentrations are
accompanied by a smaller mode diameter of the Aitken mode particles. The highest number
concentration is attributed to the frequent occurrence of NPF in these locations in pre-monsoon
(Sebastian et al., 2021b; Neitola et al., 2011). The contribution of newly formed particles to total
particles is also visible in the 75th percentile PNSDs at these sites. The number size distributions
of particles were significantly the lowest in monsoon and post-monsoon.





The median number size distribution of particles at HYD is the highest in pre-monsoon

and post-monsoon (Fig. 3e). The highest particle number concentrations in pre-monsoon and post-
monsoon can be attributed to the frequent occurrence of NPF in these seasons at the site. The
influence of NPF is also noticeable in the 75[th] percentile PNSDs. The PNSD is consistently the
lowest in monsoon, attributed to the wet scavenging of particles. The concentrations of Aitken and
accumulation mode particles are the highest in winter compared to the other seasons. The mode
diameter of PNSDs at TVM is comparatively similar in all seasons (Fig. 3f). At DEL, the mode
diameter of PNSDs is the highest in winter compared to the other seasons (Fig. 3g). The shallow
boundary layer height, stagnant atmospheric conditions, and high emission rates of aerosol
precursors in winter (Kanawade et al., 2020b) allow particles to stay close to the surface and grow
larger under high relative humidity and high condensable vapor concentrations. The median PNSD
is consistently the lowest in monsoon at TVM due to extensive wet scavenging. The strong
seasonality in PNSDs is similar to those reported earlier in India (Hooda et al., 2018; Komppula
et al., 2009; Gani et al., 2020; Kanawade et al., 2014a).

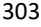

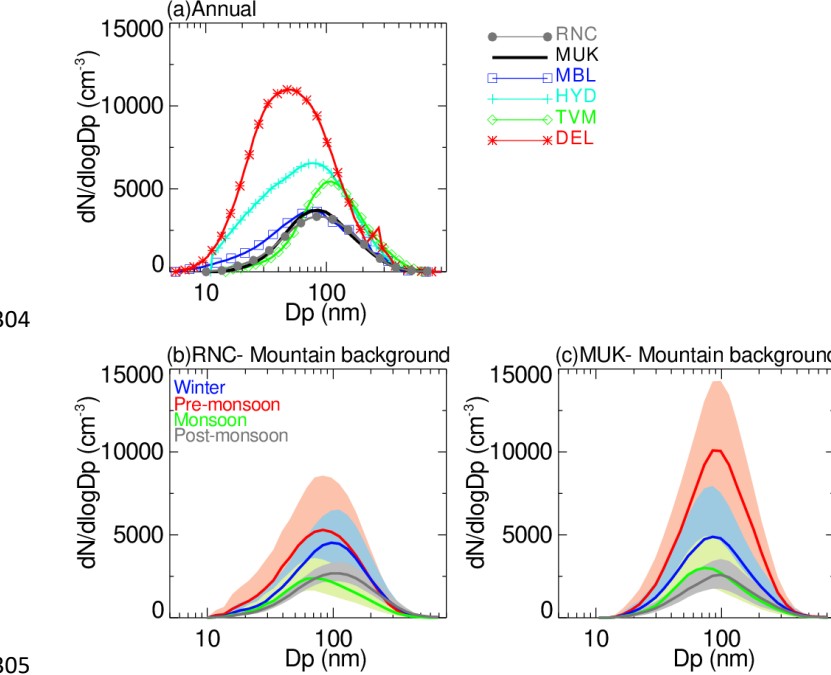



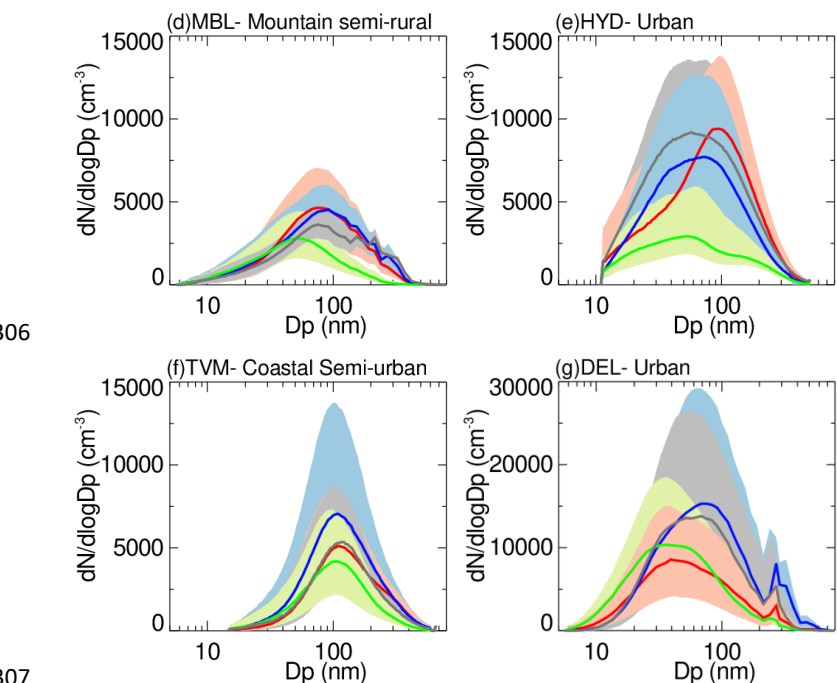



**Figure 3**. (a) Annual and (b-g) seasonal median particle number size distributions at all the sites. The solid line indicates the median, and the light-colored shading indicates 25th and 75th percentile distributions. The blue line and shading indicate winter (DJF), red line and shading indicate pre-monsoon (MAM), green line and shading indicate monsoon (JJAS), and grey line and shading indicate post-monsoon season (ON). Note that the y-axis scale is different for the DEL site.


Figure 4 shows the average observed PNSDs evolving over the day for each season, as a
contour plot, at all the sites. For RNC and MUK, the average seasonal contour plot indicates
daytime NPF in pre-monsoon. However, winter, monsoon, and post-monsoon had the lowest
concentrations of smaller particles that are not associated with NPF. For MBL, NPF occurred in
winter, pre-monsoon, and post-monsoon. For all urban sites (HYD, TVM, and DEL), the average
seasonal contour plot indicates the highest concentration of particles in morning and evening peak
traffic hours, in addition to daytime NPF. In Section 3.2, we investigate the frequency of
occurrence of NPF and its contribution to CCN concentrations.





**Figure 4**. The diurnal-seasonal median particle number size distributions at all the sites; a) Ranichauri, b) Mukteshwar, c) Mahabaleshwar, d) Hyderabad, e) Thiruvananthapuram, and f) Delhi.





Figure 5 shows the box-whisker plot of the number concentrations of Aitken,
accumulation, and total particles at all the sites. The median Aitken mode particle number
concentrations are the lowest at RNC ($1.4\times10^3$ cm$^{-3}$) and the highest at DEL ($7.1\times10^3$ cm$^{-3}$). The
median accumulation mode particle number concentrations are the lowest at MUK ($0.9\times10^3$ cm$^{-3}$)
and the highest at DEL ($2.4\times10^3$ cm$^{-3}$). The total particle number concentrations are the lowest at
MUK ($2.7\times10^3$ cm$^{-3}$) and the highest at DEL ($12.5 \times10^3$ cm$^{-3}$). The median particle number
concentrations are about 5-fold higher in urban locations (HYD, TVM, and DEL) than mountain
sites (RNC, MUK, and MBL). Overall, the size-segregated particle number concentrations show
strong spatial variability, with the lowest concentrations at the mountain sites and the highest at
the urban sites. Further, the size-segregated particle number concentrations also show the large
variability in each urban location than the mountain sites. Next, we discuss the seasonality in the
number concentration of Aitken, accumulation, and total particles in all locations to understand
space- and time-varying heterogeneity in particle number concentrations.

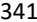

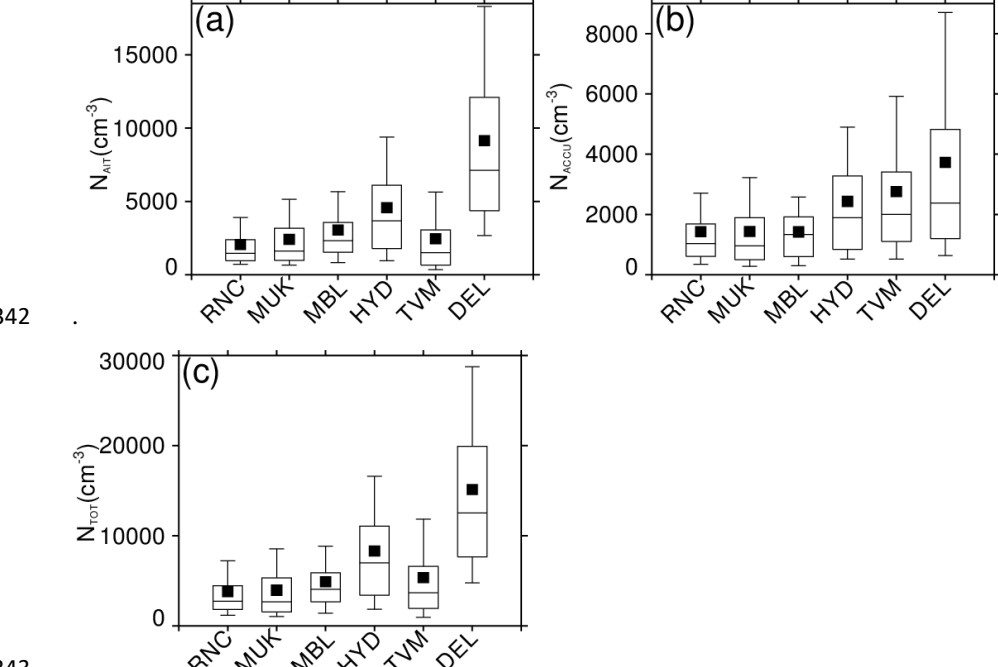

.


**Figure 5**. Box-whisker plot of the size-segregated particle number concentrations using the entire
data. The filled square indicates the mean, the horizontal line indicates the median, the top and





bottom of the box indicate 25th and 75th percentile values, and the top and bottom whiskers indicate
10th and 90th percentile values.

The histograms of the relative occurrence of Aitken mode particle number concentrations

at all the sites are presented in Figure 6(a-f). RNC and MUK show a similar seasonality in number
concentration histograms of Aitken mode particles, with a reasonably log-normal shape and the
highest concentrations in the pre-monsoon season. The lowest concentrations are observed in
monsoon and post-monsoon due to increased removal of particles by wet-scavenging. MBL does
not show notable seasonality in the number concentration histograms of Aitken mode particles.
HYD, TVM, and DEL are urban environments but show different seasonality in the number
concentration histograms of Aitken mode particles. DEL shows the highest Aitken mode particle
number concentrations in winter, and post-monsoon, TVM show the highest concentrations in
winter. In contrast, HYD shows comparable number concentrations in winter, pre-monsoon, and
post-monsoon. The highest Aitken mode number concentrations in pre-monsoon at mountain-
background sites are attributed to the high frequency of NPF occurrence in pre-monsoon (see Sect.
3.2.1). The highest Aitken mode number concentrations in winter at urban sites can be explained
by the high pre-existing particle concentration. The difference in seasonality in the number
concentration histograms of Aitken mode particles can be explained by the differences in the
atmospheric conditions (e.g., prevailing synoptic air masses, mesoscale processes such as
atmospheric boundary layer dynamics, and particle removal processes) and considerable
heterogeneity in aerosol composition (natural versus anthropogenic aerosol emission sources);
DEL is representative of a sub-tropical climate, HYD is representative of a tropical climate, and
TVM is representative of a tropical-coastal climate.


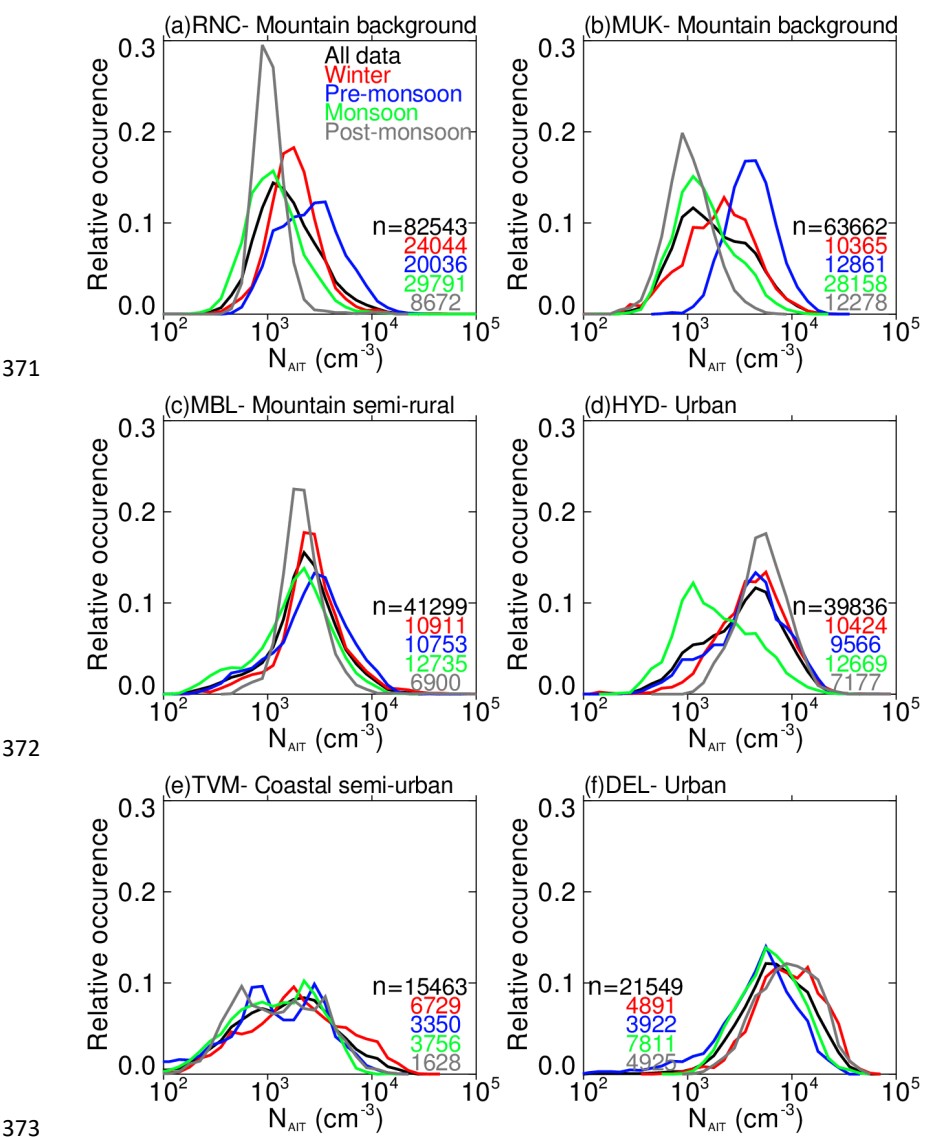




**Figure 6**. Histogram of the relative occurrence of Aitken mode particle number concentrations at the sites. The concentration bins are logarithmically spaced in the x-axis, and the y-axis shows the relative occurrence of values in each bin compared to the total number of valid observations. The thick black line indicates all data. The red, blue, green, and grey lines indicate winter (DJF), pre-monsoon (MAM), monsoon (JJAS), and post-monsoon (ON) months. n indicates the number of 10 minutes averaged valid data points.





Similar histograms of accumulation mode particles are presented in Fig. 7(a-f). The

seasonality in accumulation mode particles is slightly different as compared to Aitken mode

particles at some sites. RNC shows similar number concentration histograms of accumulation

mode particles in winter and pre-monsoon instead of dissimilar histograms for Aitken mode

particles. The number concentration histograms of accumulation mode particles at MUK are

similar to Aitken mode particles. MBL shows similar number concentration histograms in winter,

pre-monsoon, and post-monsoon, with the lowest concentrations in monsoon due to wet

scavenging. Among the urban sites, DEL shows the highest accumulation mode concentrations in

post-monsoon and winter. TVM and HYD show the highest accumulation mode concentrations in

winter and post-monsoon, respectively. The seasonality in total particles was also similar to Aitken

mode particles, indicating that Aitken mode particles constituted the most considerable fraction of

total particles at all the sites (Figure not shown). However, it is difficult to separate a fraction of

Aitken or accumulation mode particles that originated from NPF from that of the primary

emissions, especially in urban areas where the primary emission rates of aerosols are very high

(Thomas et al., 2019). The survival probability of newly formed particles to >50-100 nm size

depends on many factors such as the frequency and intensity of the NPF occurrence, availability

of condensable vapors, pre-existing particles, and atmospheric conditions. In Sect. 3.2.3, we

estimate the absolute increase of CCN concentrations from NPF following the methodology given

by Kerminen et al. (2012) and modified to remove the possible contribution from the primary

particles to CCN concentrations for any given NPF event.



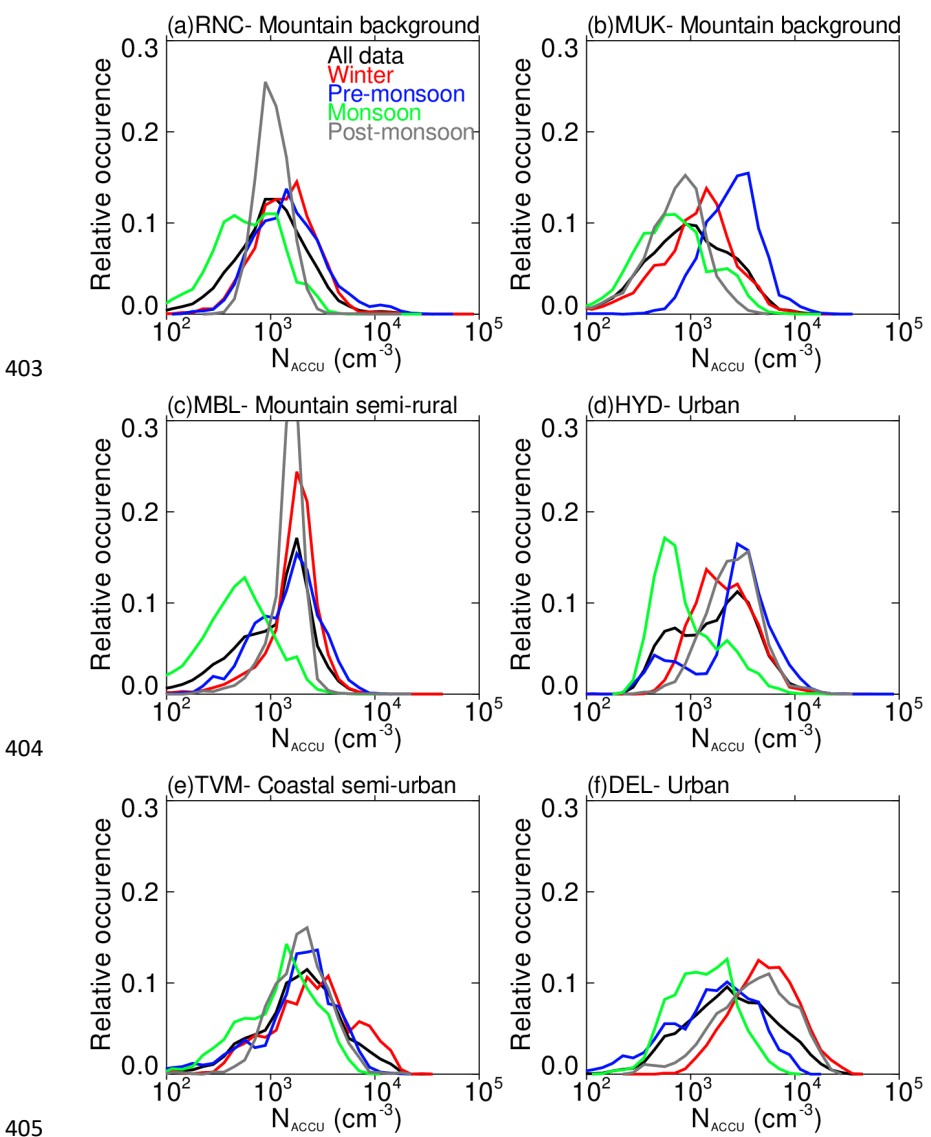

**Figure 7**. Same as Fig. 6, but for accumulation mode particle number concentrations.

**3.2 New particle formation and its contribution to CCN concentrations**

**3.2.1 NPF event characteristics**

The frequency of occurrence of NPF events, the particle formation rate of nucleation mode particles ($J_{NUC}$), and the particle growth rate of nucleation mode particles ($GR_{NUC}$) are typically derived to quantify the NPF (Kerminen et al., 2018; Nieminen et al., 2018; Kulmala et al., 2004).



These NPF characteristics are closely associated with aerosol precursor concentrations, pre-
existing aerosol particles, and atmospheric conditions. As a result, the frequency of occurrence of
NPF events varies from one location to another as well as seasonally. NPF is thought to occur
frequently during the spring (pre-monsoon) and rarely during the winter (Kanawade et al., 2012;
Dal Maso et al., 2005; Nieminen et al., 2018). However, NPF events were also observed frequently
during the thermal winter (Kulmala et al., 2004; Pikridas et al., 2012) and fall (September, October,
and November) (Rodríguez et al., 2005). These studies indicate that there is no universal pattern
in the occurrence of NPF events. Figure 8 shows the percentage of NPF, non-event, and undefined
event days based on valid observation days at all the sites. Out of a total of 586 valid observation
days at RNC, NPF events occurred on 21 days (3.9%), whereas 493 (83.7%) days were non-event
days. Out of a total of 440 valid observation days at MUK, NPF events occurred on 13 days (2.9%),
whereas 321 (73.1%) days were non-event days. Out of a total of 281 valid observation days at
MBL, NPF events occurred on 16 days (5.9%), whereas 188 (66.1%) days were non-event days.
Out of a total of 270 valid observation days at HYD, NPF events occurred on 38 days (16.3%),
whereas 124 (44.8%) days were non-event days. Out of a total of 133 valid observation days at
TVM, NPF events occurred on 23 days (16.6%), whereas 55 (41.4%) days were non-event days.
Out of a total of 139 valid observation days at DEL, NPF events occurred on 39 days (28.1%),
whereas 30 (21.1%) days were non-event days.



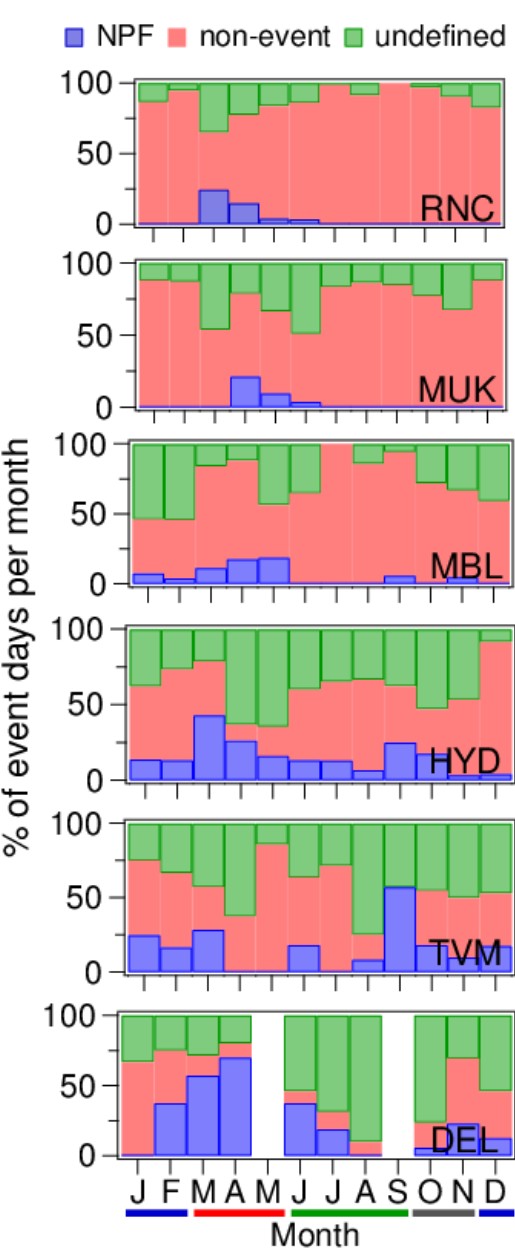

**Figure 8.** Monthly percentage of occurrence of NPF, non-event, and undefined events days based on total valid observations days at all the sites. The blue, red, green, and grey colored thick lines indicate winter, pre-monsoon, monsoon, and post-monsoon months.



### 3.2.2 Particle formation rate and growth rate

Overall, the frequency of occurrence of NPF is the highest in pre-monsoon as compared to other seasons. There is also an exception to this, with the highest frequency of NPF occurrence in the late monsoon (September) at TVM. Babu et al. (2016) have reported that NPF events over this site occurred due to a mixing of contrasting air masses due to the combined effect of mesoscale land-sea breeze circulation and local ABL dynamics. Though prevailing air masses are oceanic, the wind speeds and total rainfall were lower during September than other monsoonal months. A cleaner synoptic air mass (i.e., lower background concentrations and condensation sink), combined with the occurrence of well-defined mesoscale land-sea breeze transitions and horizontal convergence of contrasting air masses during September, was responsible for the highest NPF frequency. Amongst the sites, the mountain-background sites in the Western Himalaya (RNC and MUK) have the lowest annual mean frequency of occurrence of NPF (3.9% and 2.9%, respectively), with the highest seasonal frequency of occurrence of NPF in pre-monsoon. Previous studies also showed the infrequent occurrence of NPF at RNC (Sebastian et al., 2021b) and MUK (Neitola et al., 2011), with the highest frequency in pre-monsoon. The highest NPF frequency in pre-monsoon was connected to the planetary boundary layer uplifting to the measurement site elevation that appeared to transported aerosol precursors from nearby polluted lower-altitude regions (Hooda et al., 2018; Raatikainen et al., 2014). However, NPF occurred frequently (39%) at the Nepal Climate Observatory-Pyramid (NCO-P) site in the Eastern Himalaya (Venzac et al., 2008). A recent study also observed a very high NPF frequency (69%) at NCO-P from November to December when cleaner conditions prevailed, with little transportation from the polluted lower-altitude regions (Bianchi et al., 2021). They showed that up-valley winds bring gaseous aerosol precursors to higher altitudes. These precursors are oxidized into compounds of very low volatility and are subsequently converted into new particles during their transport to the site. The above discussion indicates that RNC and MUK mountain-background sites in the Western Himalayas are strikingly different from the NCO-P site in the Eastern Himalayas (Bianchi et al., 2021). The annual NPF frequency at RNC and MUK is lower than MBL and the high-altitude sites in Europe (Nieminen et al., 2018). DEL has the highest frequency of occurrence of NPF events in pre-monsoon (63.8%), followed by HYD (28.4%) and MBL (15.9%). TVM coastal semi-urban site witnesses frequent NPF events under the influence of continental air masses. As the air masses change from continental to mixed or marine origin, the NPF event frequency decreases (Babu et





al., 2016). NPF was also observed commonly at other urban sites in India (Kanpur and Pune) under a high source of aerosol precursors when pre-existing particle concentrations reduced sufficiently due to dilution (Kanawade et al., 2020a; Kanawade et al., 2014b). While the severe air pollution episode in Delhi in November 2016 suppressed the NPF, the co-condensation of vapors of anthropogenic origin along with water onto primary particles assisted the rapid particle growth (1.6 to 30.3 nm h$^{-1}$) (Kanawade et al., 2020b). The emission of precursor compounds from traffic and other sources in Beijing, China, also contributed significantly to the molecular cluster formation, particle growth and secondary aerosol mass formation, leading to haze formation under favorable meteorological conditions (Kulmala et al., 2021). In Europe, the atmospheric conditions (such as the solar radiation and relative humidity) appear to dictate the NPF occurrence at rural sites, whereas the increased concentrations of precursor gases are more important for the occurrence of NPF in urban areas (Bousiotis et al., 2021). This explains why NPF occurs more frequently in urban areas than rural, remote or high-altitude locations (Guo et al., 2020; Nieminen et al., 2018; Sellegri et al., 2019). This also indicates that the balance between the precursor concentration and pre-existing particles plays a vital role in the NPF occurrence. Owing to large spatial heterogeneity in aerosol precursor emissions and backgorund aerosol concentraitons in India, the chemical species contributing to aerosol nucleation and growth is unidentified (Kanawade et al., 2021). The atmospheric NPF can be quantified by calculating $J_{NUC}$ and $GR_{NUC}$ for the observed NPF events, which we discuss next.

Figure 9 shows the scatter plot of the $J_{NUC}$ and the $GR_{NUC}$ as a function of condensation sink at each site. A fairly good correlation between $J_{NUC}$ and $GR_{NUC}$ at each site (Pearson correlation coefficient of 0.48, 0.78, 0.85, 0.33, 0.68, and 0.18 at RNC, MUK, MBL, HYD, TVM, and DEL, respectively) indicates that $J_{NUC}$ and $GR_{NUC}$ are strongly coupled. The large scatter in data points is a result of important factors influencing the NPF, such as nucleation mechanisms (Dunne et al., 2016), the availability of other condensable vapors that are needed to stabilize molecular clusters containing sulfuric acid (Kirkby et al., 2011; Schobesberger et al., 2015), and atmospheric conditions (Bousiotis et al., 2021). A recent study showed that amines stabilize the nucleating cluster while organics contribute to higher concentrations of condensable vapors, particularly in urban areas (Xiao et al., 2021). The formation rate of 10 nm particles at mountain-background sites (RNC and MUK) varied from 0.01 to 0.1 cm$^{-3}$ s$^{-1}$, with a mean value of 0.08 cm$^{-3}$ s$^{-1}$. The formation rate of 5 nm particles at MBL varied from 0.02 to 0.1 cm$^{-3}$ s$^{-1}$, with a mean of





0.04 $cm^{-3}$ $s^{-1}$. The formation rate of 10 nm particles at HYD varied from 0.01 to 0.56 $cm^{-3}$ $s^{-1}$, with
a mean of 0.13 $cm^{-3}$ $s^{-1}$. The formation rate of 15 nm particles at TVM varied from 0.001 to 0.02
$cm^{-3}$ $s^{-1}$, with a mean of 0.07 $cm^{-3}$ $s^{-1}$. The formation rate of 5 nm particles at DEL varied from
0.01 to 0.5 $cm^{-3}$ $s^{-1}$, with a mean value of 0.12 $cm^{-3}$ $s^{-1}$. The mean growth rates of nucleation mode
particles during NPF events were 6.3 nm $h^{-1}$, 2.5 nm $h^{-1}$, 4.7 nm $h^{-1}$, 5.7 nm $h^{-1}$, 1.1 nm $h^{-1}$, and 3.7
nm $h^{-1}$, at RNC, MUK, MBL, HYD, TVM, and DEL, respectively. Considering all the sites, $GR_{NUC}$
during NPF events varied from 0.2 to 17.2 nm $h^{-1}$. Overall, $J_{NUC}$ and $GR_{NUC}$ are within the observed
large range of values in diverse environments in India and elsewhere (Nieminen et al., 2018;
Kerminen et al., 2018; Kulmala et al., 2004). Expectedly, the condensation sink at the start of the
NPF event is higher at urban sites than the mountain sites. The mean condensation sink at urban
sites ($16.1 \times 10^{-3}$ $s^{-}$) was twice as compared to mountain sites ($7.9 \times 10^{-3}$ $s^{-1}$). A previous study also
showed that the higher pre-existing particles at Kanpur than at Pune suppressed the particle
formation rate but favored the particle growth under high concentrations of condensable vapors
(Kanawade et al., 2014b)

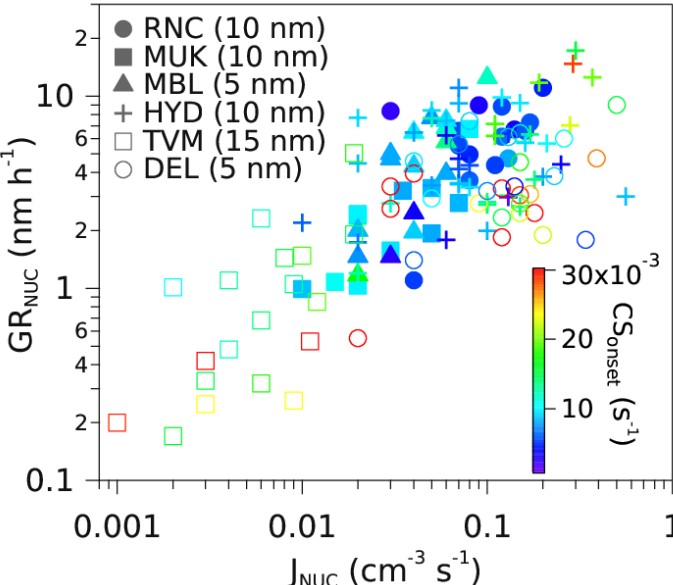


**Figure 9.** Scatter plot of the particle formation rate and the growth rate as a function of
condensation sink at each site. The condensation sink is taken at the start of the NPF event. The
lowest nucleation mode detectable size at each site is shown in the bracket.



### 3.2.3 Increase in CCN concentrations during NPF events

To reach climatologically relevant sizes, the newly formed particles must grow by condensation while avoiding coagulation removal by pre-existing particles because these freshly formed particles are small and highly diffusive (Vehkamäki and Riipinen, 2012). Based on the observed range of particle growth rates at all the sites (0.2 to 17.3 nm $h^{-1}$), newly formed particles may take from a few hours to 1-2 days to grow to CCN-active sizes (>50-100 nm). Over such time scales, it is observationally challenging to separate CCN originating from NPF from those emanating from the growth of small primary particles and direct emission of CCN-active sized particles. The increase in CCN concentrations during any given NPF event was estimated following the methodology developed by Kerminen et al. (2012), which we modified to remove CCN originating from the growth of small primary particles and direct emission of CCN-active sized particles based on non-event days.

Figure 10 shows the box-whisker plot of the absolute increase in CCN concentrations (50 and 100 nm) at all the sites. Considering all NPF events at mountain sites, increase in $CCN_{50}$ ranged from 168 $cm^{-3}$ per event to $5.2\times10^3$ $cm^{-3}$ per event, with a median value of $2.7\times10^3$ $cm^{-3}$ per event, whereas the increase in $CCN_{100}$ ranged from $0.02\times10^3$ $cm^{-3}$ per event to $1.9\times10^3$ $cm^{-3}$ per event, with the median value of $1.0\times10^3$ $cm^{-3}$ per event. The increase in $CCN_{50}$ and $CCN_{100}$ is about two-fold lower than the free tropospheric site, Chacaltaya (5240 m amsl, Bolivia), for NPF events started in the boundary layer ($5.1\times10^3$ $cm^{-3}$ per event and $1.5\times10^3$ $cm^{-3}$ per event for 50 and 100 nm, respectively) (Rose et al., 2017). The median increase in $CCN_{50}$ and $CCN_{100}$ at RNC ($2.3\times10^3$ $cm^{-3}$ per event and $0.9\times10^3$ $cm^{-3}$ per event) and MUK ($2.9\times10^3$ $cm^{-3}$ per event and $0.9\times10^3$ $cm^{-3}$ per event) are comparable to those reported at Botsalano (1420 m amsl, South Africa); $2.5\times10^3$ $cm^{-3}$ per event and $0.8\times10^3$ $cm^{-3}$ per event, respectively, but about three-fold higher than those reported at a remote continental site in Finland ($1.0\times10^3$ $cm^{-3}$ per event and $0.2\times10^3$ $cm^{-3}$ per event for 50 nm and 100 nm, respectively) (Kerminen et al., 2012). Considering all NPF events at the urban sites, $CCN_{50}$ increase ranged from $0.08\times10^3$ $cm^{-3}$ per event to $9.4\times10^3$ $cm^{-3}$ per event, with a median value of $4.3\times10^3$ $cm^{-3}$ per event, whereas $CCN_{100}$ increase ranged from $0.03\times10^3$ $cm^{-3}$ per event to $4.9\times10^3$ $cm^{-3}$ per event, with a median value of $1.2\times10^3$ $cm^{-3}$ per event. These values are about two-folds lower as compared to values reported at the station of San Pietro Capofiume, in a polluted region of the Po Valley; $7.3\times10^3$ $cm^{-3}$ per event and $2.4\times10^3$ $cm^{-3}$ per event, respectively for 50 nm and 100 nm (Laaksonen et al., 2005). The overall effect of NPF



events on the CCN concentrations at DEL was the largest since the high background number
concentrations of $CCN_{50}$ and $CCN_{100}$ resulted in a smaller relative increase, particularly in post-
monsoon and winter seasons when compared to the other sites. In order to comprehensively
investigate the atmospheric CCN budget and the contribution of NPF to it, Kerminen et al. (2012)
pointed out that the analysis should include not only NPF events but also non-event days.
Therefore, the modified methodology applied here following Kerminen et al. (2012) provides the
best representative of the increase in CCN concentrations for an NPF event.

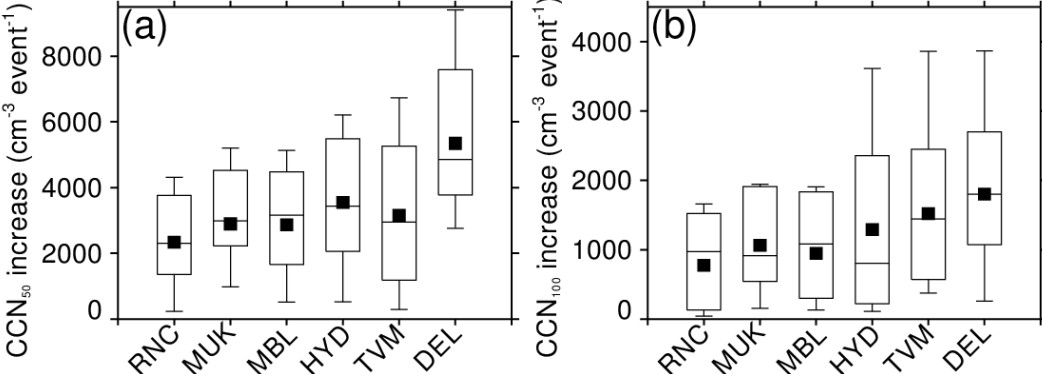

**Figure 10.** Box-whisker plot of absolute increase in CCN concentrations for (a) 50 nm and (b) 100
nm particles at all the sites based on the observed NPF and non-event events. The filled square
indicates the mean, the horizontal line indicates the median, the top and bottom of the box indicate
$25^{th}$ and $75^{th}$ percentile values, and the top and bottom whiskers indicate $10^{th}$ and $90^{th}$ percentile
values.

The sites with low pre-existing particle concentrations (hence, low condensation sink

values), high solar radiation, and cooler temperatures at high-altitude (or free tropospheric) (RNC,
MUK, and MBL) should favor NPF with enhanced frequency as compared to near-surface urban
environments (HYD, TVM, and DEL) wherein pre-existing particles concentration are high,
leading to faster removal of nucleating vapors. However, NPF in polluted environments occurs
more often than expected, with enhanced growth rates (Yu et al., 2017). Guo et al. (2014) also
reported that NPF leads to winter-time haze formation in Beijing. Kulmala et al. (2021) recently
showed that >65% of the number concentration of haze particles resulted from NPF in Beijing.





The observation sites at altitudes higher than 1000 m amsl also favored NPF at the high
condensation sinks and linked precursor gases needed to initiate nucleation and early growth
(Sellegri et al., 2019). Therefore, the low condensation sinks are not necessarily required to trigger
nucleation and early growth, provided there are high vapor production rates. Because the higher
pre-existing particle concentration is an indication of precursor-laden air, but when the
condensation sink gets very high, it inhibits aerosol nucleation. Further, at Hyderabad, about half
of the NPF events did not display aerosol nucleation (sub-3nm particle formation) with subsequent
growth of these particles to larger sizes (>10 nm), perhaps due to lower organic vapor
concentrations (Sebastian et al., 2021a). Rose et al. (2017) also reported a high frequency of NPF
occurrence for boundary layer (48%) than free troposphere (39%) conditions at Chacaltaya
mountain (5240 m amsl), Bolivia. Thus potential CCN formation was higher for NPF events
initiated in the boundary layer (67%) than free troposphere (53%). Sellegri et al. (2019) reviewed
the CCN concentrations from NPF events in the boundary layer and high-altitude locations. They
revealed that the CCN production is the highest at San Pietro Capofiume, a polluted region of the
Po Valley ($7.3 \times 10^3$ cm$^{-3}$) (Laaksonen et al., 2005) as compared to high-altitude sites (Rose et al.,
2017; Kerminen et al., 2012). Our findings are similar to these studies showing the highest increase
in CCN concentrations in urban locations (HYD, TVM, and DEL) compared to mountain locations
(RNC, MUK,  and MBL) in India. It is not possible to track the nucleated particle until it becomes
a CCN, and they are always mixed with CCN originating from primary sources. This makes it
extremely difficult to estimate CCN arising from a given NPF event. In the light of the above
discussion, these results offer some insights into potential CCN concentrations originating from
NPF.

**4 Conclusions**
In this study, we used at least one year of asynchronous particle number size distribution
measurements from six locations in India, consisting of mountain background sites (Ranichauri
and Mukteshwar), mountain rural site (Mahabaleshwar), urban sites (Delhi and Hyderabad), and
semi-urban coastal site (Thiruvananthapuram). The results from this study provide some insights
into the processes influencing particle number size distributions and CCN concentrations in
different environments (mountain and urban) of India.





We found that the regional NPF was most common in the pre-monsoon (spring) at all the
measurement sites, with an exception at TVM where NPF occurred mostly in the late monsoon
season (September), which was linked to the inflow of continental air masses that provided a
source of low volatile vapors for nucleation. During pre-monsoon, DEL has the highest frequency
of NPF occurrence (63.8%), followed by HYD (28.4%) and MBL (15.9%). NPF was the least
common during winter at all the sites, particularly at the mountain-background sites (RNC and
MUK) without a single NPF event. The high solar insolation (active photochemistry) and the
elevated boundary layer (efficient ventilation leading to low pre-existing particles near the surface)
explains the most common occurrence of NPF in the pre-monsoon (spring), but this is not a
universal NPF frequency pattern in India and elsewhere globally. We found that the $J_{NUC}$ during
NPF events tends to increase with an increasing anthropogenic influence, with an order of
magnitude higher in urban areas (0.12 $cm^{-3}$ $s^{-1}$) than mountain sites (0.06 $cm^{-3}$ $s^{-1}$). We did not find
any systematic pattern in $GR_{NUC}$, with the highest $GR_{NUC}$ at RNC (6.3 nm $h^{-1}$) and the lowest at
TVM (1.1 nm $h^{-1}$). The observed values of the NPF frequency, $J_{NUC,}$ and $GR_{NUC}$ indicate that the
regional NPF events can significantly influence the evolution of particles in the atmosphere. We
found that NPF modulates the shape of the particle number size distributions significantly,
especially at the mountain background sites (RNC and MUK), which are not directly influenced
by the local direct emissions of aerosols (traffic and industries). The number size distribution of
particles is higher in pre-monsoon at mountain-background sites, whereas it is higher in winter at
urban sites, with the exception of HYD. All sites generally show lower concentrations of particles
in monsoon due to the increased removal by wet-scavenging. The histograms of size-segregated
particle number concentrations show large variability from one site to another, reflecting the
varying contribution of different processes to the total aerosol loading. For instance, the Aitken
mode particle concentrations were the highest in pre-monsoon at mountain-background sites (RNC
and MUK), whereas they were the highest in winter at urban sites (HYD, TVM, and DEL).
Amongst the sites, the lowest measured median total particle number concentration was found in
MUK (2658 $cm^{-3}$) and the highest in DEL (12519 $cm^{-3}$).
We found that the increase in CCN concentrations during an NPF event is higher in urban
locations (4.3×10³ $cm^{-3}$ per event and 1.2×10³ $cm^{-3}$ per event for 50 nm and 100 nm, respectively)
compared to mountain-background sites (2.7×10³ $cm^{-3}$ per event and 1.0×10³ $cm^{-3}$ per event for
50 nm and 100 nm, respectively). We modified Kerminen and colleague's approach for removing



the potential contribution of primary CCN-active particles to give the best possible estimate for the increase in CCN concentrations during a given NPF event. Such analyses should be supplemented by regional model simulations or high spatial resolution measurements of NPF and CCN concentrations.

**Code availability**

Particle number size distributions data was analyzed in IGOR Pro 8.0. Figure 8 was created in IGOR Pro 8.0, while all other figures were created in IDL 8.0.

**Data availability**

Particles data will be made available upon a reasonable request to the corresponding author.

**Author contribution**:

VPK conceived the idea and designed the research. MS and VPK carried out a comprehensive data analysis. MS carried out CCN estimation analysis and interpretation with critical inputs from JRP, VV, and VPK. MS, SKK, AK, and SJ performed particle size distribution measurements and analysis. MS and VPK wrote the first draft, and MS edited with critical inputs from all co-authors.

**Competing interests**

The authors declare that they have no conflict of interest.

**Acknowledgments**

VPK was supported by the Department of Science & Technology (DST)-Science Engineering Research Board (SERB) (ECR/2016/001333) and DST-Climate Change Division Program (Aerosol/89/2017). VKS acknowledges the technical support from Sanjay Rawat for maintaining the Climate Monitoring station at Ranichauri. IITM and HACPL are fully funded by the Ministry of Earth Sciences (MoES), Govt. of India. The data collection at Thiruvananthapuram was carried out under the Aerosol Radiative Forcing over India (ARFI) project of the Indian Space Research Organisation-Geosphere Biosphere Program (ISRO-GBP). RKH, VV, EA and APH acknowledge the Academy of Finland Flagship funding (grant no. 337552). RKH and APH also acknowledge



the team of TERI, Mukteshwar and V.P. Sharma for technical support. JRP was supported by the
US Department of Energy's Atmospheric System Research, an Office of Science, Office of
Biological and Environmental Research Program, under grant DE-SC0019000.

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

Pierce, J., and Hooda, R.: New Particle Formation and Growth to Climate-Relevant Aerosols at a
Background Remote Site in the Western Himalaya, Journal of Geophysical Research:
Atmospheres, 126, 10.1029/2020JD033267, 2021b.
Sellegri, K., Rose, C., Marinoni, A., Lupi, A., Wiedensohler, A., Andrade, M., Bonasoni, P., and
Laj, P.: New Particle Formation: A Review of Ground-Based Observations at Mountain
Research Stations, Atmosphere, 10, 493, 2019.
Shika, S., Gadhavi, H., Suman, M. N. S., Ravikrishna, R., and Gunthe, S. S.: Atmospheric
aerosol properties at a semi-rural location in southern India: particle size distributions and
implications for cloud droplet formation, SN Applied Sciences, 2, 1007, 10.1007/s42452-020-
944 2804-2, 2020.

Sihto, S. L., Mikkilä, J., Vanhanen, J., Ehn, M., Liao, L., Lehtipalo, K., Aalto, P. P., Duplissy, J.,
Petäjä, T., Kerminen, V. M., Boy, M., and Kulmala, M.: Seasonal variation of CCN
concentrations and aerosol activation properties in boreal forest, Atmos. Chem. Phys., 11, 13269-
13285, 10.5194/acp-11-13269-2011, 2011.
Siingh, D., Gautam, A. S., Buchunde, P., and Kamra, A. K.: Classification of the new particle
formation events observed at a tropical site, Pune, India, Atmospheric Environment, 190, 10-22,
https://doi.org/10.1016/j.atmosenv.2018.07.025, 2018.
Singh, R. P., Dey, S., Tripathi, S. N., Tare, V., and Holben, B.: Variability of aerosol parameters
over Kanpur, northern India, 109, https://doi.org/10.1029/2004JD004966, 2004.
Srivastava, A. K., Soni, V. K., Singh, S., Kanawade, V. P., Singh, N., Tiwari, S., and Attri, S. D.:
An early South Asian dust storm during March 2012 and its impacts on Indian Himalayan
foothills: A case study, Science of The Total Environment, 493, 526-534,
https://doi.org/10.1016/j.scitotenv.2014.06.024, 2014.
Tare, V., Tripathi, S. N., Chinnam, N., Srivastava, A. K., Dey, S., Manar, M., Kanawade, V. P.,
Agarwal, A., Kishore, S., Lal, R. B., and Sharma, M.: Measurements of atmospheric parameters
during Indian Space Research Organization Geosphere Biosphere Program Land Campaign II at
a typical location in the Ganga Basin: 2. Chemical properties, 111,
https://doi.org/10.1029/2006JD007279, 2006.
Thomas, A., Sarangi, C., and Kanawade, V. P.: Recent Increase in Winter Hazy Days over
Central India and the Arabian Sea, Scientific Reports, 9, 17406, 10.1038/s41598-019-53630-3,
965 2019.

Tripathi, R. M., Khandekar, R. N., and Mishra, U. C.: Size distribution of atmospheric aerosols
in urban sites in India, Science of The Total Environment, 77, 237-244,
https://doi.org/10.1016/0048-9697(88)90059-9, 1988.
Tripathi, S. N., Tare, V., Chinnam, N., Srivastava, A. K., Dey, S., Agarwal, A., Kishore, S., Lal,
R. B., Manar, M., Kanawade, V. P., Chauhan, S. S. S., Sharma, M., Reddy, R. R., Gopal, K. R.,





Narasimhulu, K., Reddy, L. S. S., Gupta, S., and Lal, S.: Measurements of atmospheric
parameters during Indian Space Research Organization Geosphere Biosphere Programme Land
Campaign II at a typical location in the Ganga basin: 1. Physical and optical properties, 111,
https://doi.org/10.1029/2006JD007278, 2006.
Tröstl, J., Herrmann, E., Frege, C., Bianchi, F., Molteni, U., Bukowiecki, N., Hoyle, C. R.,
Steinbacher, M., Weingartner, E., Dommen, J., Gysel, M., and Baltensperger, U.: Contribution of
new particle formation to the total aerosol concentration at the high-altitude site Jungfraujoch
(3580 m asl, Switzerland), Journal of Geophysical Research: Atmospheres, 121, 11,692-611,711,
10.1002/2015jd024637, 2016.
Vehkamäki, H. and Riipinen, I.: Thermodynamics and kinetics of atmospheric aerosol particle
formation and growth, Chemical Society Reviews, 41, 5160-5173, 10.1039/C2CS00002D, 2012.
Venzac, H., Sellegri, K., Laj, P., Villani, P., Bonasoni, P., Marinoni, A., Cristofanelli, P.,
Calzolari, F., Fuzzi, S., Decesari, S., Facchini, M.-C., Vuillermoz, E., and Verza, G. P.: High
frequency new particle formation in the Himalayas, Proceedings of the National Academy of
Sciences, 105, 15666-15671, 10.1073/pnas.0801355105, 2008.
Westervelt, D. M., Pierce, J. R., and Adams, P. J.: Analysis of feedbacks between nucleation
rate, survival probability and cloud condensation nuclei formation, Atmos. Chem. Phys., 14,
5577-5597, 10.5194/acp-14-5577-2014, 2014.
Westervelt, D. M., Pierce, J. R., Riipinen, I., Trivitayanurak, W., Hamed, A., Kulmala, M.,
Laaksonen, A., Decesari, S., and Adams, P. J.: Formation and growth of nucleated particles into
cloud condensation nuclei: model–measurement comparison, Atmos. Chem. Phys., 13, 7645-
7663, 10.5194/acp-13-7645-2013, 2013.
Wiedensohler, A., Cheng, Y. F., Nowak, A., Wehner, B., Achtert, P., Berghof, M., Birmili, W.,
Wu, Z. J., Hu, M., Zhu, T., Takegawa, N., Kita, K., Kondo, Y., Lou, S. R., Hofzumahaus, A.,
Holland, F., Wahner, A., Gunthe, S. S., Rose, D., Su, H., and Pöschl, U.: Rapid aerosol particle
growth and increase of cloud condensation nucleus activity by secondary aerosol formation and
condensation: A case study for regional air pollution in northeastern China, Journal of
Geophysical Research: Atmospheres, 114, n/a-n/a, 10.1029/2008jd010884, 2009.
Wiedensohler, A., Birmili, W., Nowak, A., Sonntag, A., Weinhold, K., Merkel, M., Wehner, B.,
Tuch, T., Pfeifer, S., Fiebig, M., Fjäraa, A. M., Asmi, E., Sellegri, K., Depuy, R., Venzac, H.,
Villani, P., Laj, P., Aalto, P., Ogren, J. A., Swietlicki, E., Williams, P., Roldin, P., Quincey, P.,
Hüglin, C., Fierz-Schmidhauser, R., Gysel, M., Weingartner, E., Riccobono, F., Santos, S.,
Grüning, C., Faloon, K., Beddows, D., Harrison, R., Monahan, C., Jennings, S. G., O'Dowd, C.
D., Marinoni, A., Horn, H. G., Keck, L., Jiang, J., Scheckman, J., McMurry, P. H., Deng, Z.,
Zhao, C. S., Moerman, M., Henzing, B., de Leeuw, G., Löschau, G., and Bastian, S.: Mobility
particle size spectrometers: harmonization of technical standards and data structure to facilitate
high quality long-term observations of atmospheric particle number size distributions, Atmos.
Meas. Tech., 5, 657-685, 10.5194/amt-5-657-2012, 2012.





Xiao, M., Hoyle, C. R., Dada, L., Stolzenburg, D., Kürten, A., Wang, M., Lamkaddam, H., Garmash, O., Mentler, B., Molteni, U., Baccarini, A., Simon, M., He, X. C., Lehtipalo, K., Ahonen, L. R., Baalbaki, R., Bauer, P. S., Beck, L., Bell, D., Bianchi, F., Brilke, S., Chen, D., Chiu, R., Dias, A., Duplissy, J., Finkenzeller, H., Gordon, H., Hofbauer, V., Kim, C., Koenig, T. K., Lampilahti, J., Lee, C. P., Li, Z., Mai, H., Makhmutov, V., Manninen, H. E., Marten, R., Mathot, S., Mauldin, R. L., Nie, W., Onnela, A., Partoll, E., Petäjä, T., Pfeifer, J., Pospisilova, V., Quéléver, L. L. J., Rissanen, M., Schobesberger, S., Schuchmann, S., Stozhkov, Y., Tauber, C., Tham, Y. J., Tomé, A., Vazquez-Pufleau, M., Wagner, A. C., Wanger, R., Wang, Y., Weitz, L., Wimmer, D., Wu, Y., Yan, C., Ye, P., Ye, Q., Zha, Q., Zhou, X., Amorim, A., Carslaw, K., Curtius, J., Hansel, A., Volkamer, R., Winkler, P. M., Flagan, R. C., Kulmala, M., Worsnop, D. R., Kirkby, J., Donahue, N. M., Baltensperger, U., El Haddad, I., and Dommen, J.: The driving factors of new particle formation and growth in the polluted boundary layer, Atmos. Chem. Phys. Discuss., 2021, 1-28, 10.5194/acp-2020-1323, 2021.

Yu, F., Luo, G., Nair, A. A., Schwab, J. J., Sherman, J. P., and Zhang, Y.: Wintertime new particle formation and its contribution to cloud condensation nuclei in the Northeastern United States, Atmos. Chem. Phys., 20, 2591-2601, 10.5194/acp-20-2591-2020, 2020.

Yu, H., Ren, L., and Kanawade, V. P.: New Particle Formation and Growth Mechanisms in Highly Polluted Environments, Current Pollution Reports, 3, 245-253, 10.1007/s40726-017-0067-3, 2017.

Zhang, R., Khalizov, A., Wang, L., Hu, M., and Xu, W.: Nucleation and Growth of Nanoparticles in the Atmosphere, Chemical Reviews, 112, 1957-2011, 10.1021/cr2001756, 2012.