# Peer review of "Observations of particle number size distributions and new particle formation in six Indian locations"

_Atmospheric Chemistry and Physics, 2021_

## Referee Comment (RC2)

The study by Sebastian et al. (2021) reports asynchronous measurements of particle number size distributions (PNSD) from six stations located in contrasting environments in India. The shape of the PNSDs is first discussed, with a specific focus on the concentrations in the Aitken and accumulation modes, and the occurrence of new particle formation (NPF) in investigated in a second step. The contribution of NPF to the formation of potential cloud condensation nuclei (CCN) is finally analysed.

Although the objectives associated with this study are very interesting, I find that the methodology employed is not necessarily adequate, and, in my view, the analysis of the results could have in addition been enriched on some aspects. Therefore, I do not recommend the publication of this study in its current form. The most decisive points in my opinion are listed below.

- The first point concerns the selection of the datasets. Data availability is considered "adequate" (>60%, on what criteria?) at some sites (RNC, MUK, MBL, HYD) and limited for the others (<50% at TVM and DEL). Data availability is, in particular, very limited at TVM (34%) and one can question the relevance of the statistics that are reported for this station. Further reason to this question comes from the recent study by Rose et al (2021), who investigate the impact of reduced data availability on seasonal and annual statistics of the particle number concentration, and suggest that 50% and 60% of the data should be available to derive relevant statistics at the seasonal and annual scale, respectively.

  In addition, the data from the different sites correspond to periods that are sometimes relatively distant (between 2011 and 2020). A clear decreasing trend of the particle concentration was reported by Asmi et al. (2013) for the majority of the sites they consider in their study (located in Europe, North America, Antarctica and on Pacific Ocean islands), I therefore question the relevance of the comparative study that is made here, and which does not alert on these aspects. Further, to my knowledge, there is no study that has looked at the evolution of NPF on a global scale, and even if Nieminen et al. (2014) show that there is no clear/homogeneous trend at the boreal site of Hyytiälä over the period 1996-2012, I do not think that the authors should ignore this possibility here. To sum up, I do not think that the authors can exclude the fact that the differences observed between the sites may also be related to the selected periods, in addition to the signature of their environments. If this is not enough to completely question this study, this aspect should at least be discussed, and the following points also considered.

- In my opinion, one of the interests of a multi-site study such as this one is to be able to highlight observations common to sites with similar characteristics, or to highlight particularities, and discuss as well what explains (or may explain) the observed differences. I think that in its current form, the manuscript does not sufficiently address this last aspect. For example, the discussion at L362-368 should in my view be developed. More broadly, Section 3.1, is for me too descriptive and I find it difficult to extract a message from it. On the other hand, some additional information useful to the modeling community could easily be extracted from this analysis, such as the parameters (N, σ, d) of the representative modes of the distributions presented in Fig. 12 (similar to Asmi et al. 2011 or Rose et al. 2021); such numbers would also benefit the discussion reported at L272-312.

  Concerning the analysis of J and GR, the calculation of $J_{10}$ (with the exception of TVM, but the coverage at this site may on the other hand be too limited for such study, see previous point) and a GR on a fixed range common to all stations would have allowed a comparison of the sites between them and with the literature. Again, I find it difficult to extract a message from this analysis in its current form.

- To conclude with science, the section dedicated to the contribution of NPF to the formation of CCN also has some gaps in my opinion. I think the authors should have:
  → first recalled the main assumption that is made in this approach: particle size is considered to play a more determining role than chemical composition.
  → been clearer in the explanation of the method: for example, it is indicated "We calculated the seasonally averaged change in CCN-active particles on non-event days over the same time of day as the NPF events". What does this mean given that each event is characterized by its own start / end times? Are average start / end times considered?
  → finally, provided all the elements allowing to really evaluate the importance of NPF with respect to the (potential) CCN population at these sites: all the events certainly do not present a growth of the particles beyond 50 nm (it is at least indicated for HYD), therefore it would be interesting to know the percentage of events during which the formed particles reach a priori sizes of climatic importance, and only consider these events in the statistics reported in Fig. 10. It would also be interesting, especially for high altitude sites, to indicate the "concentration increase" observed on non-event days over the time period of interest, in order to really be able to measure the importance of NPF compared to other sources of potential CCN.

- Finally, this paper could in my view be improved in its form. For example, some lists of numbers could be replaced by tables (e.g L421-430, L495-503). Some sentences are also confusing, or have a structure that could be revised (e.g. L547-549, L575-577). Concerning Section 2.1, in particular, the information given, especially about the cities near the stations, should be homogenized (number of inhabitants missing for some). Furthermore, the reader would appreciate guidance on the impact that can be expected from these urban areas on the observations (air mass backtrajectory analysis?). More generally, a selection/reorganization of the information would often benefit the clarity of the messages.

Asmi et al.: Number size distributions and seasonality of submicron particles in Europe 2008–2009, Atmos. Chem. Phys., 11, 5505–5538, https://doi.org/10.5194/acp-11-5505-2011, 2011.

Asmi, A., Collaud Coen, M., Ogren, J. A., Andrews, E., Sheridan, P., Jefferson, A., Weingartner, E., Baltensperger, U., Bukowiecki, N., Lihavainen, H., Kivekäs, N., Asmi, E., Aalto, P. P., Kulmala, M., Wiedensohler, A., Birmili, W., Hamed, A., O'Dowd, C., G Jennings, S., Weller, R., Flentje, H., Fjaeraa, A. M., Fiebig, M., Myhre, C. L., Hallar, A. G., Swietlicki, E., Kristensson, A., and Laj, P.: Aerosol decadal trends – Part 2: In-situ aerosol particle number concentrations at GAW and ACTRIS stations, Atmos. Chem. Phys., 13, 895–916, https://doi.org/10.5194/acp-13-895-2013, 2013.

Nieminen, T., Asmi, A., Dal Maso, M., P. Aalto, P., Keronen, P., Petäjä, T., Kulmala, M. & Kerminen, V.-M. 2014: Trends in atmospheric new-particle formation: 16 years of observations in a boreal-forest environment. Boreal Env. Res. 19 (suppl. B): 191–214.

Rose et al.: Seasonality of the particle number concentration and size distribution: a global analysis retrieved from the network of Global Atmosphere Watch (GAW) near-surface observatories, Atmos. Chem. Phys., 21, 17185–17223, https://doi.org/10.5194/acp-21-17185-2021, 2021.

---

## Author Comment (AC1)

**Response to Reviewer #1 comments:**

This manuscript presents a comparison of at least 1 year of asynchronous particle size distribution (PSD) data from 6 sites across India. These include two mountain background sites (Ranichauri: Dec 2016 to Sep 2018; Mukteshwar: Jan 2012 to Dec 2013), one mountain semi-rural site (Mahabaleshwar: Mar 2015 to Mar 2016), one urban site (Hyderabad: Apr 2019 to Mar 2020), one urban coastal site (Thiruvananthapuram: Jan 2013 to Jan 2014), and one megacity (Delhi: Nov 2011 to Jan 2013). Sebastian et al. use the PSD data from the sites to compare number concentrations (Aitken mode, accumulation mode, and total number concentrations), frequency of new particle formation (NPF), and contribution of NPF to cloud condensation nuclei (CCN) concentrations among the sites. The study provides an important analysis spanning multiple sites (and all seasons) across India with implications for understanding NPF in the context of both air pollution and cloud properties. When observed, NPF contributed to higher fraction of CCN concentrations at urban sites compared to mountain (rural) sites. Overall, NPF played an important role in driving particle concentrations and size distribution.

**Response**:
We are thankful to the Reviewer for his/her suggestions and comments on our manuscript. Below, we provide a point-by-point response to comments and suggestions in the BLUE colour text. The associated modifications are shown in a red colour in the revised manuscript.

The following major changes were made to the revised manuscript.
- Figure 5 in the originally submitted manuscript was revised to reflect seasonal changes in size-segregated particle number concentrations.
- A percentage increase in $CCN_{50}$ and $CCN_{100}$ is included in Figure 10 (c and d).
- Parameters (N, σ, d) of the representative modes of the log-normal distributions are calculated and presented in the Supplement Table S1.
- The mean particle formation rate for TVM site in the originally submitted manuscript was incorrectly stated in the text as 0.07 cm$^{-3}$ s$^{-1}$, which is corrected to 0.007 cm$^{-3}$ s$^{-1}$ in revised manuscript supplement Table S2.
- Airmass trajectory analysis is presented for each site and season in the Supplement and briefly discussed in the revised manuscript Section 2.1.
- A histogram of the relative occurrence of total particles is also presented in the Supplement Figure S2.

1. Since there are relatively few PSD based studies from South Asia, this manuscript is timely and important. I have some comments which may help improve the manuscript: I think the main weakness of the analysis in its current form is that the authors compare across sites which do not have the same observed size ranges, especially in the context of the comparison of growth rates and formation rates across sites. The authors themselves write (page-10 line-241) "A direct comparison of GR and J between all of the sites is not possible because of the different size ranges covered by the instruments.". Yet, this manuscript is essentially a comparison of PNSD, $GR_{Nuc}$, and $J_{Nuc}$ between the six sites. Interestingly, the authors define $J_{LDS}$ (formation rate at the lowest detectable size) in the Methods section (Section 2.2) and never refer to it again, switching to $GR_{Nuc}$ and $J_{Nuc}$ in the Results and discussion (Section 3.2). One suggestion to provide consistent comparison across sites is to fit a multi-lognormal distribution for the particle size distributions (e.g., Hussein et al., 2005) and extrapolate for the same size range (e.g., 5–1000 nm) for each site. Then these reconstructed size

distributions can be used to compare number concentrations (then the authors can even include nucleation mode in addition to the Aitken and accumulation modes that are included in the analysis) and subsequent analysis (J, GR, etc.). [Page-9 line-225 suggests that the authors may have a mode-fitting analysis already set up.

**Response:**

Thank you for noting it. We now use $J_{LDS}$ and $GR_{LDS-25nm}$ to define the formation of the lowest detectable size (LDS) and particle growth rate between the LDS and 25 nm throughout the revised manuscript. Figure 9 shows a scatter plot of the particle formation rate ($J_{LDS}$) and the growth rate ($GR_{LDS-25nm}$) as a function of condensation sink for each site and by no means compared between the sites. Overall, each individual site shows a positive correlation between particle formation rate and growth rate.

We completely agree with the reviewers' point of view that a direct comparison between different sites is not viable because (i) the data from different sites is asynchronous, and (ii) the size distributions are not uniform. We are comparing particle data between seasons at respective sites and do not intend to compare particle data between sites even though they have been plotted on the same figure (Figure 3, 6 and 7). In effect, we have restructured/removed some sentences in the results and discussion section.

This study essentially covers the time period from 2011 to 2019 when considering all sites. We have calculated yearly averaged particle volume size distributions in the size range from 0.1 to 1.0 μm for four sites in India where more than five years of AERONET data is available (Gandhi College, Jaipur, Kanpur and Pune) (Figure R1). Gandhi College is a typical semi-urban type, Pune and Kanpur are typically urban, and Jaipur is a mixed urban semi-arid environment. We avoided the year 2020 due to nationwide lockdown owing to COVID-19, which reduced primary anthropogenic emissions. There is no clear linear increasing trend in particle volume size distributions in the size range from 0.1 to 1.0 μm, while several studies found a significant rise in anthropogenic aerosol loading over India (Dey and Di Girolamo, 2011; Krishna Moorthy et al., 2013; Ramachandran et al., 2012; Thomas et al., 2019). The averaged particle volume size distribution over the entire period show large variability for particles larger than 0.3 μm. The calculated trend in total volume concentration in the size range from 0.1 to 1 μm also shows an insignificant increasing trend at all sites over the time period from 2011 to 2019 (Figure R2). Considering all the sites, the total volume concentration changed from -6% to 14%. From this analysis, it can be concluded that particle volume size distribution properties in the size range from 0.1 to 1 μm may not have changed drastically over the study time period. Similar trends and variability can be applied to sites considered in this study. It may also be noted that a variety of factors can influence trends in aerosols like urbanization, meteorology and regional climate. Nevertheless, we do not exclude the rising trends in aerosols over India. Still, we have refrained from comparing NPF characteristics between the sites.

[Figure]

**Figure R1**. Yearly averaged particle volume size distribution in the size range from 0.1 to 1 μm during 2011-2019 (colored lines) and the mean particle volume size distribution with standard deviation (black line) based on AERONET observations at (a) Gandhi College, (b) Jaipur, (c) Kanpur, and (d) Pune

[Figure]

**Figure R2**. The trend in yearly averaged total volume concentration in the size range of 0.1-1.0 μm during 2011-2019 at Gandhi College, Jaipur, Kanpur and Pune. The dotted line shows the linear fit line, and the slope of the linear fit is given in the legend.

As suggested by the Reviewer, we have used (Hussein et al., 2005) to extrapolate the PNSDs down to 5 nm to have uniform particle number size distribution data across all the sites. The multi log-normal distribution was fitted through measured particle number size distributions to extrapolate the data down to 5 nm. However, we show that multi log-normal fitted PNSD down to 5 nm deviates from the actual value (Fig. R3). We used measurements down to 5.6 nm from two sites, Delhi and Mahabaleshwar. For these sites, the extrapolated PNSD using a multi log-normal fitting largely deviates from the actual value during a typical NPF event observed on 22 June 2012, while the extrapolated PNSD fitted well to the measured PNSD during a typical non-event observed on 24 June 2012. The same exercise was performed for the Mahabaleshwar site for typical NPF and non-event days. It can be concluded from this analysis that a multi log-normal distribution fitting fails to capture/identify/detect the particle modes below ~15 nm on NPF event days. The NPF characteristics calculation is subjective, and estimation based on fitted PNSDs will further augment errors in the computation of NPF characteristics e.g., nucleation mode number concentration, particle formation rate and growth rate. Therefore, we abstain from extrapolating the PNSDs for the size range for which it was not measured.

[Figure]

**Figure R3**. Measured PNSD in the size range 5.6-1050 nm (black line connected by a plus sign), fitted multi log-normal distribution in the same size range (blue line), and fitted multi log-normal distribution in the size range 14.7-1150 nm (red line) during an (a) NPF event on 22 June 2012, (b) non-event on 24 June 2012, at Delhi (c) NPF event on 21 March 2016 and (d) non-event on 25 July 2015, at Mahabaleshwar.

2. The manuscript should include instrumentation setup details (including inlet and sampling tubing information) for each site or refer to previous articles from these sites which contain this information.
   **Response:**
   We have included instrumentation setup details for each site where applicable or cited relevant references which contain instrumentation setup details as indicated below.

For MUK, the reference has been cited as "More details of the site and aerosol sampling can be found in Hyvärinen et al. (2009)".

For Delhi, the following details included - "The WRAS system uses a stainless-steel inlet tube with an integrated Nafion drier to dry the aerosol sample. A detailed description of the site and aerosol sampling is given elsewhere (Jose et al., 2021)."

For MBL, the following discussion is added, and the paper has already been cited. "The WRAS has a stainless-steel inlet tube with a an integrated Nafion dryer to reduce the relative humidity to ~40%."

For TVM, the following details are included in the revised manuscript: "The ambient air was sampled from a height of 3 m above ground level through a manifold inlet fitted with $PM_{10}$ size cut impactor at 16.67 LPM flow rate. Subsequently, the flow was distributed among various aerosol instruments connected with electrically conductive tubing. To restrict high relative humidity conditions, a diffusion dryer (Make: TSI, Model: 3062) employing silica gel was used."

3. Since the datasets range across a decade (Delhi 2011-2012 to Hyderabad 2019-2020), it may be helpful to present the dates in the figures (or caption) where the comparison across sites is presented. This will help put the comparison in the context of not only the different sites, but also different years as presumably most of these sites have become more polluted over the last decade. Furthermore, a brief discussion on the possible implications of the changes in particle size distribution over the past decade will be helpful.
**Response:**
Table 1 summarizes measurement details, including time periods, instrument, particle size range and time resolution. To help the reader, we have added a statement in each figure caption. "Note that measurements are from different time periods for each site (refer to Table 1)"

4. The definition of seasons (Table 2) warrants some discussion. For example, why use "pre-monsoon" and not "spring" and "summer". Furthermore, "monsoon" spans across four months for all sites. What are the implications of the season definitions to the summary results (when averaging using these periods) given the differences in climatology for each site? To be clear, I am not asking the authors to necessarily change the season definitions, just to justify and discuss their implications on the results.
**Response:**
The classification is based on India Meteorological Department (IMD). December, January and February are the coldest months in a year at all these sites (Winter). March through May are the warmest months at all the sites (summer or pre-monsoon). We referred to it as pre-monsoon as per the IMD definition. The onset of monsoon happens on the southern tip of India in early June and encompasses the entire country by mid-August, and the retreat phase lasts till the end of September. Therefore four months are considered as the monsoon season as per IMD definition. October and November months consist of the post-monsoon season. Also, Table 2 clearly identifies the meteorological characteristics of each season.

5. The discussion on precursors in this manuscript seems to be primarily based on existing literature. Is it possible to include some approximate quantitative comparison of precursor concentrations across the six sites (and by season), perhaps using $SO_2$ data (if available) to

calculate H2$_S$O$_4$ proxy (Dada et al., 2020)?

**Response:**

Unfortunately, we do not have measurements of precursor gases (such as SO$_2$, organics etc.) for these sites except Hyderabad. Our recent study calculated sulfuric acid proxy based on SO$_2$ concentrations in Hyderabad (Sebastian et al., 2021).

Since "primary" and "secondary" is now routinely used in the context of mass-spectra derived source apportionment, the authors should be intentional and clear while using the terms "primary" and "secondary" in the context of the PSD-based NPF analysis presented throughout the manuscript (e.g., on page 10, lines 255).

**Response:**

The "primary" and "secondary" spectra are used to separate between primary and secondary mass (regardless of whether secondary mass condenses to a particle that came from primary emission). We do not classify the aerosols based on mass concentration. Here, the likely "primary' and "secondary" sources of particle number concentrations are indicated.

6. The analysis on the "relative occurrence of Aitken mode and accumulation mode" (pages 16-19, including Figures 6 and 7) is not clear to me. To my knowledge this is not a standard analysis and requires more/clearer context and guidance for the reader to understand the results and their interpretations. For example, in the context of Figure 6 (x-axis: Aitken mode concentration; y-axis: "relative occurrence"), the authors write "a reasonably log-normal shape…" (page-16 line 351). Perhaps I am missing something, but I am unable to understand this discussion.

   **Response:**

   Relative occurrence explains how frequently is a particular number concentration occurs for a particle mode. The relative occurrence of size-segregated particle number concentrations is presented to find the maximum relative occurrence of a particular particle size range (type) in different seasons to infer possible causes of variability on relative occurrence. For instance, higher Aitken mode particle number concentrations in pre-monsoon (March through May) than other seasons indicate the potential contribution from NPF processes and are in line with the highest NPF frequency in the pre-monsoon season. Delhi has the highest occurrence of Aitken mode particles during the winter season, indicating the dominance of anthropogenic sources and conducive meteorological conditions (Kanawade et al., 2020). The term 'reasonably log-normal distribution' is removed to avoid confusion for the reader.

7. (Page 25 line 506) "Expectedly, the condensation sink at the start of the NPF event is higher at urban sites than the mountain sites. The mean condensation sink at urban sites (16.1×10$^{-3}$ s-) was twice as compared to mountain sites (7.9×10$^{-3}$ s$^{-1}$)." What do "start of NPF event" condensation sink values mean? Are they averaged over a few minutes or hours? In the second sentence, what is the averaging period for the "mean condensation sink"?

   **Response:**

   The condensation sink at the start of the event is taken as one-hour average CS just before the start of the NPF event. It has been clearly stated in the Figure 9 caption.

8. When presenting CCN increase, the authors should consider also including the fraction increase (%) over the "baseline" in addition to the magnitude increase (cm$^{-3}$) which the authors have done (in abstract and conclusions as well).

   **Response:**

   We have included percentage change in CCN$_{50}$ and CCN$_{100}$ increase in the revised manuscript as suggested by the Reviewer as shown below Figure R4.

[Figure]

**Figure R4.** Box-whisker plot of percentage increase in CCN concentrations for (c) 50 nm and (d) 100 nm particles at all the sites based on the observed NPF and non-event events.

9. Please use full caption in Figure 7 (should be able to stand alone).
   **Response:**
   Included the full caption in Figure 7.

10. Updated ACP/Copernicus guidelines state "it is important that the colour schemes used in your maps and charts allow readers with colour vision deficiencies to correctly interpret your findings." This means that jet/rainbow color scales need to be changed to other appropriate color scales. More here: https://www.atmospheric-chemistry-and-physics.net/submission.html#figurestables
    **Response:** We have checked figures for colour vision deficiencies, and it seems to be appropriately reflected.

**References:**

Dada, L., Ylivinkka, I., Baalbaki, R., Li, C., Guo, Y., Yan, C., Yao, L., Sarnela, N., Jokinen, T., Daellenbach, K. R., Yin, R., Deng, C., Chu, B., Nieminen, T., Wang, Y., Lin, Z., Thakur, R. C., Kontkanen, J., Stolzenburg, D., Sipilä, M., Hussein, T., Paasonen, P., Bianchi, F., Salma, I., Weidinger, T., Pikridas, M., Sciare, J., Jiang, J., Liu, Y., Petäjä, T., Kerminen, V.-M., and Kulmala, M.: Sources and sinks driving sulfuric acid concentrations in contrasting environments: implications on proxy calculations, 20, 11747–11766, https://doi.org/10.5194/acp-20-11747-2020, 2020.

Hussein, T., Dal Maso, M., Petäjä, T., Koponen, I., Paatero, P., Aalto, P., Hämeri, K., and Kulmala, M.: Evaluation of an automatic algorithm for fitting the particle number size distribution, Boreal Environment Research, 10, 337–355, 2005.
**Citation**: https://doi.org/10.5194/acp-2021-804-RC1

---

## Author Comment (AC2)

**Response to Reviewer #2 comments:**

The study by Sebastian et al. (2021) reports asynchronous measurements of particle number size distributions (PNSD) from six stations located in contrasting environments in India. The shape of the PNSDs is first discussed, with a specific focus on the concentrations in the Aitken and accumulation modes, and the occurrence of new particle formation (NPF) in investigated in a second step. The contribution of NPF to the formation of potential cloud condensation nuclei (CCN) is finally analysed.
Although the objectives associated with this study are very interesting, I find that the methodology employed is not necessarily adequate, and, in my view, the analysis of the results could have in addition been enriched on some aspects. Therefore, I do not recommend the publication of this study in its current form. The most decisive points in my opinion are listed below.

**Response**:
We are thankful to the Reviewer for his/her suggestions and comments on our manuscript. Below, we provide a point-by-point response to comments and suggestions in the BLUE colour text. The associated modifications are shown in a red colour in the revised manuscript.

The following major changes were made to the revised manuscript.
- Figure 5 in the originally submitted manuscript was revised to reflect seasonal changes in size-segregated particle number concentrations.
- A percentage increase in $CCN_{50}$ and $CCN_{100}$ is included in Figure 10 (c and d).
- Parameters (N, σ, d) of the representative modes of the log-normal distributions are calculated and presented in Supplement Table S1.
- The mean particle formation rate for TVM site in the originally submitted manuscript was incorrectly stated in the text as $0.07 \text{ cm}^{-3} \text{ s}^{-1}$, which is corrected to $0.007 \text{ cm}^{-3} \text{ s}^{-1}$ in the revised manuscript supplement Table S2.
- Airmass trajectory analysis is presented for each site and season in the Supplement and briefly discussed in the revised manuscript Section 2.1.
- A histogram of the relative occurrence of total particles is also presented in the Supplement Figure S2.

1. The first point concerns the selection of the datasets. Data availability is considered "adequate" (>60%, on what criteria?) at some sites (RNC, MUK, MBL, HYD) and limited for the others (<50% at TVM and DEL). Data availability is, in particular, very limited at TVM (34%) and one can question the relevance of the statistics that are reported for this station. Further reason to this question comes from the recent study by Rose et al (2021), who investigate the impact of reduced data availability on seasonal and annual statistics of the particle number concentration, and suggest that 50% and 60% of the data should be available to derive relevant statistics at the seasonal and annual scale, respectively.

In addition, the data from the different sites correspond to periods that are sometimes relatively distant (between 2011 and 2020). A clear decreasing trend of the particle concentration was reported by Asmi et al. (2013) for the majority of the sites they consider in their study (located in Europe, North America, Antarctica and on Pacific Ocean islands), I therefore question the relevance of the comparative study that is made here, and which does not alert on these aspects. Further, to my knowledge, there is no study that has looked at the

evolution of NPF on a global scale, and even if Nieminen et al. (2014) show that there is no clear/homogeneous trend at the boreal site of Hyytiälä over the period 1996-2012, I do not think that the authors should ignore this possibility here. To sum up, I do not think that the authors can exclude the fact that the differences observed between the sites may also be related to the selected periods, in addition to the signature of their environments. If this is not enough to completely question this study, this aspect should at least be discussed, and the following points also considered.

**Response**:

In the originally submitted manuscript, we have clearly stated that the data availability is adequate at four sites while it is limited at two sites (TVM and DEL) for readers to relate the statistics derived from these sites. Figure R1 shows the bar plot of seasonal data availability for all the sites. DEL has lower data availability during winter, while TVM has lower data availability during pre-monsoon, monsoon and post-monsoon seasons. Rose et al. (2021) used data availability of >50% so that more data can be used for the analysis as opposed to Laj et al. (2020), which used data availability of >75%. (Rose et al., 2021) indicated that the criterion of >50% is not estimated on strict statistical analysis. Though the data availability is lower for TVM and DEL, these measurements are very useful to understand the general features in the data and can be used with caution, especially the regions for which such analyses are rarely reported.

[Figure]

**Figure R1**. Seasonal data availability for all the sites.

We completely agree with the reviewers' point of view that a direct comparison between different sites is not viable because (i) the data from different sites is asynchronous, and (ii) the size distributions are not uniform. We are comparing particle data between seasons at respective sites and do not intend to compare particle data between them even though they have been plotted on the same figure (Figure 3, 6 and 7). In effect, we have restructured some sentences in the results and discussion section.

This study essentially covers the time period from 2011 to 2019 when considering all sites. We have calculated yearly averaged particle volume size distributions in the size range from 0.1 to 1.0 μm for four sites in India where more than five years of AERONET data is available (Gandhi College, Jaipur, Kanpur and Pune) (Figure R2). Gandhi College is a typical semi-urban type, Pune and Kanpur are typically urban, and Jaipur is a mixed urban semi-arid environment. We avoided the year 2020 due to nationwide lockdown owing to COVID-19, which reduced primary anthropogenic emissions. There is no clear linear increasing trend in particle volume size distributions in the size range from 0.1 to 1.0 μm, while several studies found a significant rise in anthropogenic aerosol loading over India (Dey and Di Girolamo, 2011; Krishna Moorthy et al., 2013; Ramachandran et al., 2012; Thomas et al., 2019). The averaged particle volume size distribution over the entire period show large variability for

particles larger than 0.3 µm. The calculated trend in total volume concentration in the size range from 0.1 to 1 µm also shows an insignificant increasing trend at all sites over the time period from 2011 to 2019 (Figure R3). Considering all the sites, the total volume concentration changed from -6% to 14%. From this analysis, it can be concluded that particle volume size distribution properties in the size range from 0.1 to 1 µm may not have changed drastically over the study time period. Similar trends and variability can be applied to sites considered in this study. It may also be noted that a variety of factors can influence trends in aerosols like urbanization, meteorology and regional climate. Nevertheless, we do not exclude the rising trends in aerosols over India. Still, we have refrained from comparing NPF characteristics between the sites.

In the originally submitted manuscript, Figure 9 shows a scatter plot of the particle formation rate and the growth rate as a function of condensation sink for each site and by no means compared between the sites. Overall, each site shows a positive correlation between the formation rate and growth rate.

[Figure]

**Figure R2**. Yearly averaged particle volume size distribution in the size range from 0.1 to 1 µm during 2011-2019 (colored lines) and the mean particle volume size distribution with standard deviation (black line) based on AERONET at (a) Gandhi College, (b) Jaipur, (c) Kanpur, and (d) Pune

[Figure]

**Figure R3**. The trend in yearly averaged total volume in the size range of 0.1-1.0 μm during 2011-2019 at Gandhi College, Jaipur, Kanpur and Pune. The dotted line shows the linear fit line, and the slope of the linear fit is given in the legend.

In my opinion, one of the interests of a multi-site study such as this one is to be able to highlight observations common to sites with similar characteristics, or to highlight particularities, and discuss as well what explains (or may explain) the observed differences. I think that in its current form, the manuscript does not sufficiently address this last aspect. For example, the discussion at L362-368 should in my view be developed. More broadly, Section 3.1, is for me too descriptive and I find it difficult to extract a message from it. On the other hand, some additional information useful to the modelling community could easily be extracted from this analysis, such as the parameters (N, σ, d) of the representative modes of the distributions presented in Fig. 12 (similar to Asmi et al. 2011 or Rose et al. 2021); such numbers would also benefit the discussion reported at L272-312.

Concerning the analysis of J and GR, the calculation of J10 (with the exception of TVM, but the coverage at this site may on the other hand be too limited for such study, see previous point) and a GR on a fixed range common to all stations would have allowed a comparison of the sites between them and with the literature. Again, I find it difficult to extract a message from this analysis in its current form.

**Response:**
We agree with the Reviewer's point of view on a multi-site study to be able to highlight observations common to sites with similar characteristics, but we refrained from comparing observations from similar sites as the period of study is different for different sites. The discussions at L362-368 point to the possible reasons for the seasonality in particle number concentrations. Major findings from other studies related to seasonality in particle number concentrations across these sites are discussed in the manuscript.

Section 3.1 mostly deals with explaining the size distribution characteristics across all the sites. We believe that the section adequately discusses the seasonality in particle number size distributions and number concentrations in size ranges common to all sites (Aitken and Accumulation mode) for all the sites.

As suggested by the Reviewer, we have calculated N, σ and Dp values for the particle number size distributions for all the sites on a seasonal basis. The values are tabulated in Table R1. The table is included in the supplementary information as Table S1.

Table R1. Parameters of the modes identified for the description of median particle number size distributions from all six sites shown in Figure 3. N, Dp and σ are the number concentration, geometric mean diameter and the standard deviation of the distribution.

| | Unimodal | | | Bimodal | | | | | |
|---|---|---|---|---|---|---|---|---|---|
| Site | N | Dp | σ | $N_1$ | $Dp_1$ | $σ_1$ | $N_2$ | $Dp_2$ | $σ_2$ |
| **Annual** | | | | | | | | | |
| RNC | 2555 | 87.5 | 2.0 | 591 | 49.3 | 1.9 | 1963 | 101.8 | 1.9 |
| DEL | 9670 | 50.9 | 2.2 | 8237 | 44.9 | 2.0 | 1465 | 121.3 | 1.9 |
| HYD | 6401 | 63.4 | 2.5 | 2097 | 27.3 | 1.8 | 4186 | 90.0 | 1.9 |
| MBL | 3166 | 74.2 | 2.3 | 3104 | 72.9 | 2.2 | 48 | 197.3 | 1.2 |
| MUK | 2573 | 85.5 | 1.9 | 301 | 65.1 | 1.5 | 2276 | 90.5 | 1.9 |
| TVM | 3463 | 111.2 | 1.8 | 3379 | 109.6 | 1.8 | 85 | 330.8 | 1.3 |
| **Winter** | | | | | | | | | |
| RNC | 3205 | 94.6 | 1.9 | 876 | 53.0 | 1.9 | 2357 | 109.2 | 1.8 |
| DEL | 13555 | 68.7 | 2.3 | 12878 | 65.9 | 2.1 | 678 | 298.0 | 1.2 |
| HYD | 7314 | 61.1 | 2.3 | 3165 | 33.9 | 1.8 | 3990 | 95.2 | 1.8 |
| MBL | 3817 | 84.4 | 2.3 | 4877 | 100.2 | 2.6 | 789 | 319.4 | 0.6 |
| MUK | 3374 | 86.0 | 1.9 | 3344 | 85.5 | 1.9 | 28 | 256.0 | 1.2 |
| TVM | 4437 | 113.2 | 1.8 | 4266 | 110.6 | 1.8 | 169 | 320.0 | 1.3 |
| **Pre-Monsoon** | | | | | | | | | |
| RNC | 4012 | 81.2 | 2.0 | 2721 | 64.6 | 1.9 | 1280 | 118.7 | 1.7 |
| DEL | 7708 | 49.8 | 2.3 | 4622 | 35.8 | 1.9 | 3093 | 96.0 | 2.1 |
| HYD | 7726 | 82.0 | 2.2 | 1858 | 24.5 | 1.8 | 6007 | 98.7 | 1.8 |
| MBL | 3702 | 78.6 | 2.1 | 5034 | 100.5 | 2.4 | 1342 | 228.3 | 0.5 |
| MUK | 6488 | 91.1 | 1.8 | 1748 | 62.5 | 1.9 | 4760 | 101.4 | 1.7 |
| TVM | 3241 | 122.4 | 1.8 | 2933 | 115.3 | 1.7 | 282 | 313.0 | 1.3 |
| **Monsoon** | | | | | | | | | |
| RNC | 1774 | 78.4 | 2.0 | 85 | 58.3 | 1.3 | 1693 | 81.1 | 2.0 |
| DEL | 9336 | 40.2 | 2.2 | 5059 | 27.4 | 1.9 | 4194 | 66.8 | 1.9 |
| HYD | 3141 | 49.2 | 2.8 | 2844 | 45.4 | 2.5 | 210 | 196.9 | 1.4 |
| MBL | 2187 | 50.3 | 2.1 | 1960 | 47.8 | 2.3 | 255 | 58.9 | 1.4 |
| MUK | 1984 | 79.4 | 1.9 | 1765 | 73.9 | 1.7 | 223 | 199.5 | 1.5 |
| TVM | 2603 | 103.4 | 1.8 | 1565 | 93.5 | 2.1 | 1109 | 110.1 | 1.6 |
| **Post-Monsoon** | | | | | | | | | |

| | | | | | | | | | |
|---|---|---|---|---|---|---|---|---|---|
| RNC | 2072 | 102.0 | 2.0 | 441 | 52.0 | 1.8 | 1629 | 118.7 | 1.9 |
| DEL | 12152 | 60.6 | 2.2 | 11881 | 59.5 | 2.1 | 286 | 263.9 | 1.1 |
| HYD | 9949 | 58.7 | 2.5 | 9335 | 57.5 | 2.5 | 123 | 157.5 | 1.3 |
| MBL | 3277 | 88.5 | 2.4 | 2937 | 79.4 | 2.2 | 289 | 237.6 | 1.3 |
| MUK | 1782 | 93.7 | 1.9 | 1743 | 93.3 | 2.0 | 50 | 99.0 | 1.2 |
| TVM | 3176 | 117.5 | 1.8 | 3099 | 116.2 | 1.7 | 86 | 360.7 | 1.3 |

We now use $J_{LDS}$ and $GR_{LDS-25nm}$ to define the formation of the lowest detectable size and particle growth rate between the LDS and 25 nm for respective sites. $GR_{LDS-25nm}$ and $J_{LDS}$ for each site have been plotted in Figure 9 for visualizing the overall association between $GR_{LDS-25nm}$ and $J_{LDS}$ when considering all sites. We understand that the GR and J values cannot be made for a fixed range as the LDS is different for each site.

2.  To conclude with science, the section dedicated to the contribution of NPF to the formation of CCN also has some gaps in my opinion. I think the authors should have:

   ➤ first recalled the main assumption that is made in this approach: particle size is considered to play a more determining role than chemical composition.
   ➤ been clearer in the explanation of the method: for example, it is indicated "We calculated the seasonally averaged change in CCN-active particles on non-event days over the same time of day as the NPF events". What does this mean given that each event is characterized by its own start / end times? Are average start / end times considered?
   ➤ finally, provided all the elements allowing to really evaluate the importance of NPF with respect to the (potential) CCN population at these sites: all the events certainly do not present a growth of the particles beyond 50 nm (it is at least indicated for HYD), therefore it would be interesting to know the percentage of events during which the formed particles reach a priori sizes of climatic importance, and only consider these events in the statistics reported in Fig. 10. It would also be interesting, especially for high altitude sites, to indicate the "concentration increase" observed on non-event days over the time period of interest, in order to really be able to measure the importance of NPF compared to other sources of potential CCN.

   **Responses:**

   ➤ We have edited the statement as "In typical ambient in-cloud supersaturations, the total number of particles from 50 nm to >100 nm can be considered as a proxy for CCN concentrations assuming fixed chemical composition."
   ➤ We have addded statements in the revised mansucript as "The start of the NPF event is the time when nucleation mode particle number concentrations increase rapidly during an NPF event." and "For non-event days, the seasonally averaged start of the NPF event time was chosen to calculate $N_{CCNprior}$. $N_{CCNmax}$ on non-event days was taken similar to NPF event days, as a maximum one-hour average concentration of particles larger than 50 nm (and 100 nm)."
   ➤ All identified NPF events have particle mode diameter growing beyond 50 nm. Only these events are used for calculating the values plotted in Figure 10. The time evolutions of seasonally averaged diurnal particle number size distributions (Figure 4) show that particles grow beyond 100 nm at all sites, with the exception of monsoon season at some sites. The red open circles show an absolute increase in CCN concentrations for 50 nm and

100 nm on non-event days (second term in Eq. 2) in the Fig. R4.

[Figure]

**Figure R4.** Box-whisker plot of absolute increase in CCN concentrations for (a) 50 nm and (b) 100 nm particles on NPF event days (First team in Eq. 2). The filled square box indicates the mean, the horizontal line indicates the median, the top and bottom of the box indicate 25th and 75th percentiles values and the top and bottom of the whiskers indicate the 10th and 90th percentile values. The red open circles show the mean CCN concentrations for (a) 50 nm and (b) 100 nm on non-event days (Second term in Eq. 2).

3. Finally, this paper could in my view be improved in its form. For example, some lists of numbers could be replaced by tables (e.g L421-430, L495-503). Some sentences are also confusing, or have a structure that could be revised (e.g. L547-549, L575-577). Concerning Section 2.1, in particular, the information given, especially about the cities near the stations, should be homogenized (number of inhabitants missing for some). Furthermore, the reader would appreciate guidance on the impact that can be expected from these urban areas on the observations (air mass back trajectory analysis?). More generally, a selection/reorganization of the information would often benefit the clarity of the messages.

**Response**
Agree, and we have included a table summarising the frequency of occurrence of NPF events and non-NPF events, $GR_{LDS-25nm}$ and $J_{LDS}$ in Supplement as Table S2 as shown below in Table R2.

Table R2. Number of valid observation days, number of NPF days (percentage), number of non-NPF days (percentage), mean formation rates and mean growth of particles at all six sites of study.

| Site code | valid observation days | NPF days | Non-event days | $J_{LDS}$ $(cm^{-1}s^{-1})$ | $GR_{LDS-25\ nm}$ $(nm\ h^{-1})$ |
|---|---|---|---|---|---|
| RNC | 586 | 21 (3.9%) | 493 (83.7%) | 0.11±0.05 | 6.3±2.4 |
| MUK | 440 | 13 (2.9%) | 321 (73.1%) | 0.04±0.02 | 2.5±1.6 |
| MBL | 281 | 16 (5.9%) | 188 (66.1%) | 0.04±0.02 | 4.7±3.0 |
| HYD | 270 | 38 (16.3%) | 124 (44.8%) | 0.13±0.11 | 5.7±3.6 |
| TVM | 133 | 23 (16.6%) | 55 (41.4%) | 0.007±0.005 | 1.1±1.1 |
| DEL | 139 | 39 (28.1%) | 30 (21.1%) | 0.13±0.10 | 3.7±2.1 |

Lines 547-549 is rephrased as

"High background number concentrations of $CCN_{50}$ and $CCN_{100}$ in Delhi resulted in a smaller relative increase of CCN from NPF, during post-monsoon and winter seasons when compared to the other sites."

Lines 575-577 is rephrased as

"The high pre-existing particle concentration is also an indication of precursor-laden air. But when the condensation sink gets very high, it inhibits aerosol nucleation."

The airmass history is analysed using HYSPLIT transport model (Fig. R5). The seasonal trajectory density plots for all six sites are added to the supplementary as Figure S1.

[Figure]

**Figure R5.** HYSPLIT modelled 72-hour backward air mass back trajectory density starting at 500 m for (a) Ranichauri, (b) Mukteshwar, (c) Mahabaleshwar, (d) Hyderabad, (e) Thiruvananthapuram and (f) Delhi.

**References**

Asmi et al.: Number size distributions and seasonality of submicron particles in Europe 2008–2009, Atmos. Chem. Phys., 11, 5505–5538, https://doi.org/10.5194/acp-11-5505-2011, 2011.

Asmi, A., Collaud Coen, M., Ogren, J. A., Andrews, E., Sheridan, P., Jefferson, A., Weingartner, E., Baltensperger, U., Bukowiecki, N., Lihavainen, H., Kivekäs, N., Asmi, E., Aalto, P. P., Kulmala, M., Wiedensohler, A., Birmili, W., Hamed, A., O'Dowd, C., G Jennings, S., Weller, R., Flentje, H., Fjaeraa, A. M., Fiebig, M., Myhre, C. L., Hallar, A. G., Swietlicki, E., Kristensson, A., and Laj, P.: Aerosol decadal trends – Part 2: In-situ aerosol particle number concentrations at GAW and ACTRIS stations, Atmos. Chem. Phys., 13, 895–916, https://doi.org/10.5194/acp-13-895-2013, 2013.

Nieminen, T., Asmi, A., Dal Maso, M., P. Aalto, P., Keronen, P., Petäjä, T., Kulmala, M. & Kerminen, V.- M. 2014: Trends in atmospheric new-particle formation: 16 years of observations in a boreal-forest environment. Boreal Env. Res. 19 (suppl. B): 191–214.

Rose et al.: Seasonality of the particle number concentration and size distribution: a global analysis retrieved from the network of Global Atmosphere Watch (GAW) near-surface observatories, Atmos. Chem. Phys., 21, 17185–17223, https://doi.org/10.5194/acp-21-17185-2021, 2021.

Dey, S., Di Girolamo, L., 2011. A decade of change in aerosol properties over the Indian subcontinent. 38.

Hussein, T., Dal Maso, M., Petäjä, T., Koponen, I., Paatero, P., Aalto, P., Hämeri, K., Kulmala, M., 2005. Evaluation of an automatic algorithm for fitting the particle number size distribution. Boreal Environment Research 10, 337-355.

Hyvärinen, A.P., Lihavainen, H., Komppula, M., Sharma, V.P., Kerminen, V.-M., Panwar, T., Viisanen, Y., 2009. Continuous measurements of optical properties of atmospheric aerosols in Mukteshwar, northern India. Journal of Geophysical Research-Atmospheres 114.

Jose, S., Mishra, A.K., Lodhi, N.K., Sharma, S.K., Singh, S., 2021. Characteristics of Aerosol Size Distributions and New Particle Formation Events at Delhi: An Urban Location in the Indo-Gangetic Plains. 9.

Kanawade, V.P., Srivastava, A.K., Ram, K., Asmi, E., Vakkari, V., Soni, V.K., Varaprasad, V., Sarangi, C., 2020. What caused severe air pollution episode of November 2016 in New Delhi? Atmospheric Environment 222, 117125.

Kerminen, V.-M., Paramonov, M., Anttila, T., Riipinen, I., Fountoukis, C., Korhonen, H., Asmi, E., Laakso, L., Lihavainen, H., Swietlicki, E., Svenningsson, B., Asmi, A., Pandis, S.N., Kulmala, M., Petäjä, T., 2012. Cloud condensation nuclei production associated with atmospheric nucleation: a synthesis based on existing literature and new results. Atmos. Chem. Phys. 12, 12037-12059.

Krishna Moorthy, K., Suresh Babu, S., Manoj, M.R., Satheesh, S.K., 2013. Buildup of aerosols over the Indian Region. 40, 1011-1014.

Laj, P., Bigi, A., Rose, C., Andrews, E., Lund Myhre, C., Collaud Coen, M., Lin, Y., Wiedensohler, A., Schulz, M., Ogren, J.A., Fiebig, M., Gliß, J., Mortier, A., Pandolfi, M., Petäja, T., Kim, S.W., Aas, W., Putaud, J.P., Mayol-Bracero, O., Keywood, M., Labrador, L., Aalto, P., Ahlberg, E., Alados Arboledas, L., Alastuey, A., Andrade, M., Artíñano, B., Ausmeel, S., Arsov, T., Asmi, E., Backman, J., Baltensperger, U., Bastian, S., Bath, O., Beukes, J.P., Brem, B.T., Bukowiecki, N., Conil, S., Couret, C., Day, D., Dayantolis, W., Degorska, A., Eleftheriadis, K., Fetfatzis, P., Favez, O., Flentje, H., Gini, M.I., Gregorič, A., Gysel-Beer, M., Hallar, A.G., Hand, J., Hoffer, A., Hueglin, C., Hooda, R.K., Hyvärinen, A., Kalapov, I., Kalivitis, N., Kasper-Giebl, A., Kim, J.E., Kouvarakis, G., Kranjc, I., Krejci, R., Kulmala, M., Labuschagne, C., Lee, H.J., Lihavainen, H., Lin, N.H., Löschau, G., Luoma, K., Marinoni, A., Martins Dos Santos, S., Meinhardt, F., Merkel, M., Metzger, J.M., Mihalopoulos, N., Nguyen, N.A., Ondracek, J., Pérez, N., Perrone, M.R., Petit, J.E., Picard, D., Pichon, J.M., Pont, V., Prats, N., Prenni, A., Reisen, F., Romano, S., Sellegri, K., Sharma, S., Schauer, G., Sheridan, P., Sherman, J.P., Schütze, M., Schwerin, A., Sohmer, R., Sorribas, M., Steinbacher, M., Sun, J., Titos, G., Toczko, B., Tuch, T., Tulet, P., Tunved, P., Vakkari, V., Velarde, F., Velasquez, P., Villani, P., Vratolis, S., Wang, S.H., Weinhold, K., Weller, R., Yela, M., Yus-Diez, J., Zdimal, V., Zieger, P., Zikova, N., 2020. A global analysis of climate-relevant aerosol properties retrieved from the network of Global Atmosphere Watch (GAW) near-surface observatories. Atmos. Meas. Tech. 13, 4353-4392.

Ramachandran, S., Kedia, S., Srivastava, R., 2012. Aerosol optical depth trends over different regions of India. Atmospheric Environment 49, 338-347.

Rose, C., Collaud Coen, M., Andrews, E., Lin, Y., Bossert, I., Lund Myhre, C., Tuch, T., Wiedensohler, A., Fiebig, M., Aalto, P., Alastuey, A., Alonso-Blanco, E., Andrade, M., Artíñano, B., Arsov, T., Baltensperger, U., Bastian, S., Bath, O., Beukes, J.P., Brem, B.T., Bukowiecki, N., Casquero-Vera, J.A., Conil, S., Eleftheriadis, K., Favez, O., Flentje, H., Gini, M.I., Gómez-Moreno, F.J., Gysel-Beer, M., Hallar, A.G., Kalapov, I., Kalivitis, N., Kasper-Giebl, A., Keywood, M., Kim, J.E., Kim, S.W., Kristensson, A., Kulmala, M., Lihavainen, H., Lin, N.H., Lyamani, H., Marinoni, A., Martins Dos Santos, S., Mayol-Bracero, O.L., Meinhardt, F., Merkel, M., Metzger, J.M., Mihalopoulos, N., Ondracek, J., Pandolfi, M., Pérez, N., Petäjä, T., Petit, J.E., Picard, D., Pichon, J.M., Pont, V., Putaud, J.P., Reisen, F., Sellegri, K., Sharma, S., Schauer, G., Sheridan, P., Sherman, J.P., Schwerin, A., Sohmer, R., Sorribas, M., Sun, J., Tulet, P., Vakkari, V., van Zyl, P.G., Velarde, F., Villani, P., Vratolis, S., Wagner, Z., Wang, S.H., Weinhold, K., Weller, R., Yela, M., Zdimal, V., Laj, P., 2021. Seasonality of the particle number concentration and size distribution: a global analysis retrieved from the network of Global Atmosphere Watch (GAW) near-surface observatories. Atmos. Chem. Phys. 21, 17185-17223.

Sebastian, M., Kanawade, V.P., Pierce, J.R., 2021. Observation of sub-3nm particles and new particle formation at an urban location in India. Atmospheric Environment 256, 118460.

Thomas, A., Sarangi, C., Kanawade, V.P., 2019. Recent Increase in Winter Hazy Days over Central India and the Arabian Sea. Scientific Reports 9, 17406.

Westervelt, D.M., Pierce, J.R., Riipinen, I., Trivitayanurak, W., Hamed, A., Kulmala, M., Laaksonen, A., Decesari, S., Adams, P.J., 2013. Formation and growth of nucleated particles into cloud condensation nuclei: model–measurement comparison. Atmos. Chem. Phys. 13, 7645-7663.

---

## Author Response (AR2)

**Response to Editors comments:**

**Scientific comment:**

Low GR (below 1 nm h-1) and low J (below 0.02 cm-3 s-1) are almost exclusively from TVM, which has the highest detection limit (15 nm). The time scale for growth is 50 h, 20 h, 10 h at the rate of 0.2 nm-1, 0.5 nm h-1 and 1 nm h-1, respectively. Such slow GR is typically only observed at very homogenous and sparsely populated Arctic / Antarctic environments. Considering coastal environment with land-sea breeze (time scale of 6 h or so) and emissions from 1 M population in the region, this result is surprising to me. Any comments or speculation, why the GR is so slow?

**Response**:

Note that "Lowest Detectable Size" (LDS) is replaced by "Smallest Detectable Size (SDS)" throughout the manuscript, including Fig. 9 as "smallest" goes well with size than "lowest"

There are two possibilities why GR is so slow/low from TVM. First, the CS is relatively higher in TVM (similar to a megacity like Delhi, as seen in Fig.9), which indicates a higher probability of condensable vapors lost on pre-existing particles, which may contribute to lower GR, and thus lower $J_{SDS}$. Secondly,  in theory, the steepness of GR curve increases with decreasing particle size, since TVM, with the highest SDS among all the sites, has the lowest steepness of GR curve between SDS and 25 nm resulting lower GR and this lower $J_{SDS}$.

**Technical comments:**

Line 655: I would use less significant figures for the highest number concentrations.

**Response**: Done

Figure 5. Please consider logarithmic scale in the y-axis.

**Response**: Done.

Figure 6&7. To my eye, the black line is no thicker than any of the other. In Figure 7c, the grey curve goes outside the axis limits.

**Response**: In Figures 6&7, all lines have equal thickness. The figure caption is updated as "The black, blue, red, green, and grey lines indicate all data, winter (DJF), pre-monsoon (MAM), monsoon (JJAS), and post-monsoon (ON), respectively". We changed the y-axis range to show a grey curve within the plotting window.

**Response to Reviewer's comments:**

I think that this revised article incorporates the bulk of the comments that the two reviewers made to the initially submitted version. The authors also convincingly addressed some of the comments that they cannot incorporate in the edited manuscript. However, there are some comments that the authors have incompletely addressed:

1) Since the analysis on the "relative occurrence of Aitken mode and accumulation mode" (pages 18-21; Fig 6 and Fig 7) is not a standard analysis, I recommend providing some context for the reader. Perhaps a couple of sentences around line 377 when you introduce the concept.

**Response**: We have introduced the concept as "The relative occurrence of the number concentrations of size-segregated (Aitken and accumulation) particles was calculated to determine the maximum concentrations of a given particle mode in different seasons at all sites."

Technical corrections:

2) The authors should change the colorscales in figures 1, 4, and 9 (and now Fig S1 as well) to a uniform colorscale. The Copernicus guidelines (https://www.atmospheric-chemistry-and-physics.net/submission.html#figurestables) mentions: "For more information on the background and importance of addressing this issue, we refer to Stoelzle & Stein (2021) (https://hess.copernicus.org/articles/25/4549/2021/)." The cited article literally discusses why "rainbow colormap" (color scheme in question) is misleading. I understand that the aerosol community has used the jet/rainbow colorscale for decades and it is inconvenient for us to change from the "standard" rainbow/jet colorscale. But, given the guidelines, I have to point this out.

**Response**: The corresponding author is sorry to have missed incorporating this in the earlier revision. We have changed color schemes for figures 1, 4, 9, and S1.

3) Units missing in Table S1.

**Response**: Specified.

Overall, the article is much improved and should be published subject to technical corrections.